# RinRK1 enhances NF receptors accumulation in nanodomain-like structures at root-hair tip

Ning Zhou[1,2], Xiaolin Li[1,2], Zhiqiong Zheng[1], Jing Liu[1], J. Allan Downie [3] & Fang Xie[1] ✉

Legume-rhizobia root-nodule symbioses involve the recognition of rhizobial Nod factor (NF) signals by NF receptors, triggering both nodule organogenesis and rhizobial infection. *RinRK1* is induced by NF signaling and is essential for infection thread (IT) formation in *Lotus japonicus*. However, the precise mechanism underlying this process remains unknown. Here, we show that RinRK1 interacts with the extracellular domains of NF receptors (NFR1 and NFR5) to promote their accumulation at root hair tips in response to rhizobia or NFs. Furthermore, Flotillin 1 (Flot1), a nanodomain-organizing protein, associates with the kinase domains of NFR1, NFR5 and RinRK1. RinRK1 promotes the interactions between Flot1 and NF receptors and both RinRK1 and Flot1 are necessary for the accumulation of NF receptors at root hair tips upon NF stimulation. Our study shows that RinRK1 and Flot1 play a crucial role in NF receptor complex assembly within localized plasma membrane signaling centers to promote symbiotic infection.

The plasma membrane (PM), which forms the boundary of living cells and serves as their interface with the external environment, controls the transport of molecules into and out of cells and the transmission of external signals into the cell interior. The PM is a two-dimensional lipid bilayer containing integral or associated proteins in which the lateral flow and distribution of lipids and proteins are highly heterogeneous. This leads to the formation of dynamic protein clusters and PM subcompartments with varying shapes and sizes[1–3]. The assembly of dynamic protein clusters in the PM (i.e., nanodomains or microdomains) increases the stability and activity of the membrane proteins and protein complexes. Various signaling molecules that activate plant PM receptors and nanodomains are potential triggers for the assembly of specific receptor complexes and signaling outputs[4]. For example, in *Arabidopsis thaliana* roots, the brassinosteroid (BR) receptor BRI1 (Brassinosteroid Insensitive 1) is localized in different PM nanodomains and this is crucial for BR signal transmission[5]. Live-cell imaging unveiled that FLS2 (Flagellin Sensing 2) and BRI1 establish discrete PM nanodomains characterized by specific remorin proteins. This spatial

segregation allows for the separation of FLS2 and BRI1 from their respective signaling components within dedicated PM nanodomains and serves as a regulatory mechanism for plant immune responses or growth[6].

The symbiosis between legumes and rhizobia enables plants to directly take up nitrogen fixed by the bacterial microsymbionts. Legumes secrete flavonoids and related compounds that are perceived by rhizobia leading to the production of strain-specific lipochito-oligosaccharide signals called Nodulation factors (NFs). The NFs are perceived by legume LysM-type receptor-like kinases (RLKs), including NFR1 (Nod Factor Receptors 1) and NFR5 in *Lotus japonicus* and their counterparts, NFP (Nod Factor Perception) and LYK3 (Lysine Motif Kinase 3) in *Medicago truncatula*[7–10]. The site of production of NFs is important in determining where infections occur. Individual rhizobia can preferentially attach to root-hair tips via the binding of surface polysaccharides to a lectin localized at the root-hair tip[11]. The localized production of NFs induces root hairs to curl back on themselves entrapping the rhizobia, which multiplies to form a microcolony[12], a

[1]National Key Laboratory of Plant Molecular Genetics, CAS Center for Excellence in Molecular Plant Sciences, Shanghai Institute of Plant Physiology and Ecology, Chinese Academy of Sciences, Shanghai, China. [2]University of the Chinese Academy of Sciences, Beijing, China. [3]John Innes Centre, Norwich Research Park, NR4 7UH Norwich, UK. ✉e-mail: fxie@cemps.ac.cn

process that further increases localized NF production. At this point, the root hair starts to produce an infection thread (IT), which is essentially an intracellular tunnel down which the rhizobia can grow[13,14].

SPFH (stomatin/prohibitin/flotillin/HflK/C) domain proteins and remorins act as scaffold proteins in the PM nanodomain in plant cells and have been widely used as nanodomain markers[2,15–18]. SPFH proteins are evolutionarily conserved across prokaryotes and eukaryotes and have been identified in various plant species, including *Arabidopsis*, rice, and legumes. The first legume flotillin-like protein, GmNOD53b, was isolated from soybean (*Glycine max*), and GmNOD53b-like proteins have been identified in the plant-made peri-bacteroid membrane that surrounds nitrogen-fixing rhizobia in pea (*Pisum sativum*) nodules[19,20]. In *M. truncatula*, Flot2 and Flot4, which are induced by rhizobia, play an essential role in the initiation and elongation of IT within root hair during interactions with rhizobia[21]. Subsequent investigations revealed that rhizobia can trigger the translocation of MtLYK3-GFP to nanodomains and enhance the co-localization of MtLYK3 and MtFlot4 at root-hair tips. The localization of MtFlot4 in this nanodomain is abolished in *hcl-1*, which carries a mutated *MtLYK3*, resulting in deficient kinase activity[22]. Remorins serve as unique markers for membrane-associated nanodomains in plants. *SymREM1* (Symbiotic Remorin 1) is induced by NFs in the roots of both *M. truncatula* and *L. japonicus*. SymREM1 interacts with the NF receptors LjNFR1/MtLYK3 and LjNFR5/MtNFP, as well as with the leucine-rich repeat (LRR)-type symbiosis receptor kinase (LjSymRK/MtDMI2), to enhance rhizobial infections. It has been observed in vitro that both LjNFR1 and LjSymRK can phosphorylate LjSymREM1[23,24]. MtSymREM1 is recruited to nanodomains in a MtFlot4-dependent manner, suggesting that MtFlot4 is critical for the assembly of primary nanodomains[25]. A similar pattern was observed for the soybean transmembrane and nanodomain-localized protein GmFWL1 (FW2-2-like 1), which interacts with GmFlot2/4 and may have scaffolding functions[26]. The lateral stability of nanodomains and the nucleation points during de novo nanodomain assembly, such as actin or microtubules, are required for the accumulation of proteins in membrane nanodomains[27]. The microtubule-localized protein MtDREPP, which can interact with MtFlot4 and MtSymREM1, is re-localized into symbiosis-specific membrane nanodomains in a stimulus-dependent manner[28].

In *L. japonicus*, NFR1 and NFR5 form a protein complex that recognizes NFs. The *nfr1* and *nfr5* mutants are unresponsive to *M. loti* and purified NFs and are unable to mediate *M. loti* infections of the epidermis and cortical cell nodule organogenesis[7,8,29]. A recent study of NF receptors revealed that their intracellular kinase domains alone are capable of initiating the signaling pathway to promote organogenesis, whereas successful infection necessitates functional ectodomains[30]. Hence, the NF signal generated by NFR1 and NFR5 splits into two pathways. One pathway involves nuclear $Ca^{2+}$ spiking and is mediated by the common symbiosis genes that are essential and sufficient for nodule organogenesis. The other pathway is associated with the NF receptors and is required for the progression of the infection process via ITs[31]. RinRK1 (Rhizobia infection Receptor-like Kinase 1) encodes a LRR-type RLK that is induced by NFs and is required for IT-mediated rhizobial infection in *L. japonicus*. Alignment of the kinase domain of RinRK1 with other RLKs showed the absence of several conserved amino acid residues crucial for kinase activity, and in vitro kinase assay further confirmed the lack of autophosphorylation activity in RinRK1[32,33].

In this study, we show that RinRK1 interacts with both NFR1 and NFR5 through their extracellular domains. Furthermore, we identify the interaction between Flot1 and the kinase domains of NFR1, NFR5, and RinRK1. RinRK1 further promotes the interaction of Flot1 with NFR1/NFR5 and, along with Flot1, is required for their accumulation at the root-hair tips in response to rhizobia or NF treatment.

These findings provide compelling evidence that RinRK1 facilitates the assembly of a protein complex involving NFR1, NFR5, and Flot1 within nanodomain-like structures and enables IT formation.

## Results

### RinRK1 and NF receptors interact through their extracellular domains

We speculated that RinRK1 could associate with NF receptors, leading to the promotion of IT formation in root hairs. To investigate this, we tested whether RinRK1 can interact with NFR1 and NFR5 by using a split-ubiquitin yeast two-hybrid (Y2H) system to examine the potential interactions between the full-length proteins. The yeast grew well on a selective medium supplemented with 3-amino-1,2,4-triazole (3-AT) when RinRK1-ubiquitin was co-expressed with ubiquitin-linked NFR1 or NFR5, indicating that they can interact. LjSymREM1, which is known to interact with NFR1[24], served as the positive control. MtLYK10, an ortholog of the EPS receptor LjEPR3, which functions independently of NF receptors in symbiotic infection[34], was used as a negative control (Fig. 1a). To identify which domains are important for these interactions, the extracellular and intracellular domains of these proteins were co-expressed in yeast using the GAL4 Y2H assay. The presence of the extracellular domains (EDs), but not the cytoplasmic domains (CDs), of RinRK1 and NFR1 or NFR5 enabled yeast growth on the selective medium. The protein expression in yeast was confirmed by western blot analysis (Fig. 1b and Supplementary Fig. 1). These results suggest that the interaction occurs between extracellular LysM domains of NFR1 and NFR5 and the LRR domain of RinRK1.

The interactions between RinRK1 and NFR1 or NFR5 were validated using in vivo co-immunoprecipitation (Co-IP). Full-length RinRK1 tagged with Flag (RinRK1-Flag) was co-expressed with Myc-tagged NFR1 (NFR1-Myc), NFR5 (NFR5-Myc), or with MtLYK10 (LYK10-Myc), in *Nicotiana benthamiana* leaves. The immunoprecipitated assay indicated that RinRK1 can interact with NFR1 and NFR5, while no interaction was detected with MtLYK10 (Fig. 1c). To verify the necessity of the RinRK1 extracellular domain for interactions with NFR1 and NFR5, a construct encoding only the RinRK1 N-terminal LRR and its transmembrane domain (RinRK1-N) was used in the Co-IP assay. The results showed that RinRK1-N was able to associate with NFR1-Myc and NFR5-Myc (Fig. 1c), indicating that the RinRK1 extracellular domain is crucial for these interactions.

Moreover, split-luciferase complementation and Bimolecular fluorescence complementation (BiFC) assays were conducted in *N. benthamiana* and confirmed that NFR1 and NFR5, but not MtLYK10, can interact with RinRK1 in vivo (Fig. 1d and Supplementary Fig. 2). We conclude that RinRK1 can interact with NFR1 and NFR5 through their extracellular domains.

### RinRK1 accumulates at root-hair tips in response to *M. loti* NFs

Multiple proteome analyses using detergent-resistant membranes (DRMs), a commonly used method for studying membrane lipid rafts, have revealed the abundance of RLKs in PM nanodomains[35,36]. Additionally, the RinRK1 homologs, At5g16590[37,38] and At2g26730[39], have been detected in the DRM fraction of *A. thaliana*. Therefore, we hypothesized that RinRK1 might exhibit a similar behavior. To verify this, we expressed RinRK1-eGFP in *N. benthamiana* leaves, isolated detergent-insoluble PM fractions, and detected RinRK1 in this fraction (Fig. 2a), indicating its presence in nanodomain-like structures.

We then investigated RinRK1 subcellular localization and dynamics in *L. japonicus* root hairs using wild-type (Gifu) hairy roots expressing p*Ub:RinRK1-eGFP* before and after inoculation with *Mesorhizobium loti* strain R7A. It was confirmed that the expression of p*Ub:RinRK1-eGFP* in *rinrk1-1* roots complemented the infection defect (Supplementary Fig. 3a, b), indicating the fusion protein is functional. RinRK1-eGFP dynamics in root hairs were analyzed using live-cell confocal imaging. In the absence of *M. loti*, RinRK1-eGFP

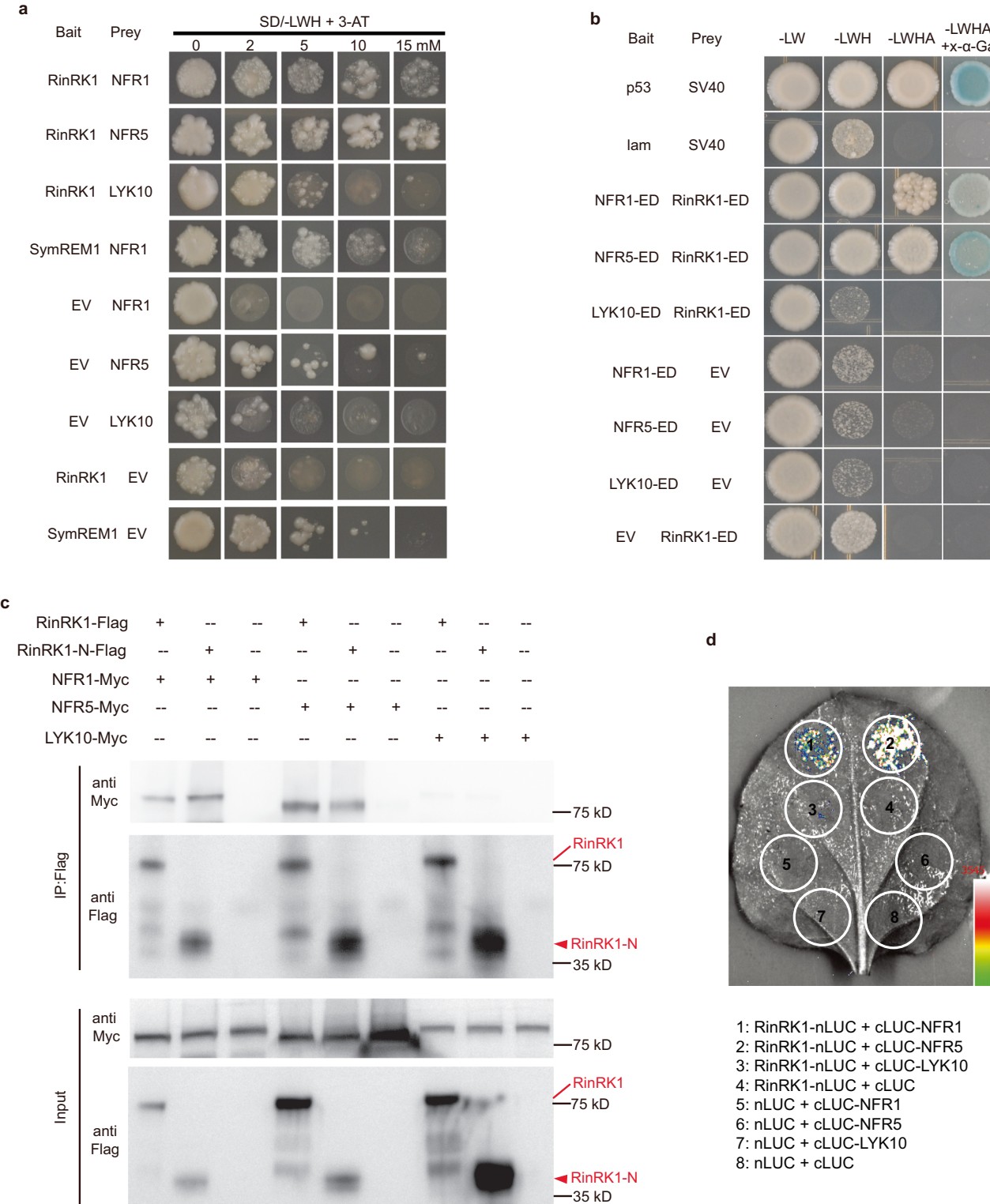

**Fig. 1 | RinRK1 interacts with NFR1 and NFR5 via their extracellular domains.**
**a** The split-ubiquitin yeast two-hybrid system was used to analyze the interaction between full-length RinRK1 and NFR1, NFR5, SymREM1, and MtLYK10. Yeast cells were first grown on SD-Leu-Trp medium and then transferred to SD-Leu-Trp-His (SD-LWH) medium supplemented with 3-amino-1,2,4-triazole (3-AT) to suppress endogenous His biosynthesis. The SymREM1- NFR1 interaction served as the positive control. **b** The GAL4 yeast two-hybrid system was used to analyze potential interactions between the extracellular domain (ED) of RinRK1 and NFR1, NFR5, and MtLYK10. SV40 and p53 served as the positive control, while SV40 and lam were used as the negative control. **c** Co-immunoprecipitation analysis of the interactions between the full-length RinRK1 (RinRK1-Flag) or the RinRK1 N-terminal

(RinRK1-N-Flag) and NFR1, NFR5, or MtLYK10 in *N. benthamiana* leaves. The plasmids containing RinRK1-Flag or RinRK1-N-Flag were co-expressed with Myc-tagged NFR1, NFR5, or MtLYK10 in *N. benthamiana* leaves. Co-IP was conducted using anti-Flag agarose, and the precipitated proteins were detected by immunoblot analysis with anti-Flag and anti-Myc antibodies. **d** Split-luciferase complementation assay of the interactions between the full-length RinRK1 and NFR1, NFR5, or MtLYK10 in *N. benthamiana* leaves. The indicated constructs were co-expressed in *N. benthamiana* leaves, and luciferase complementation imaging was conducted two days after agroinfiltration. Fluorescence signal intensity is indicated. One representative example out of three independent experiments is shown.

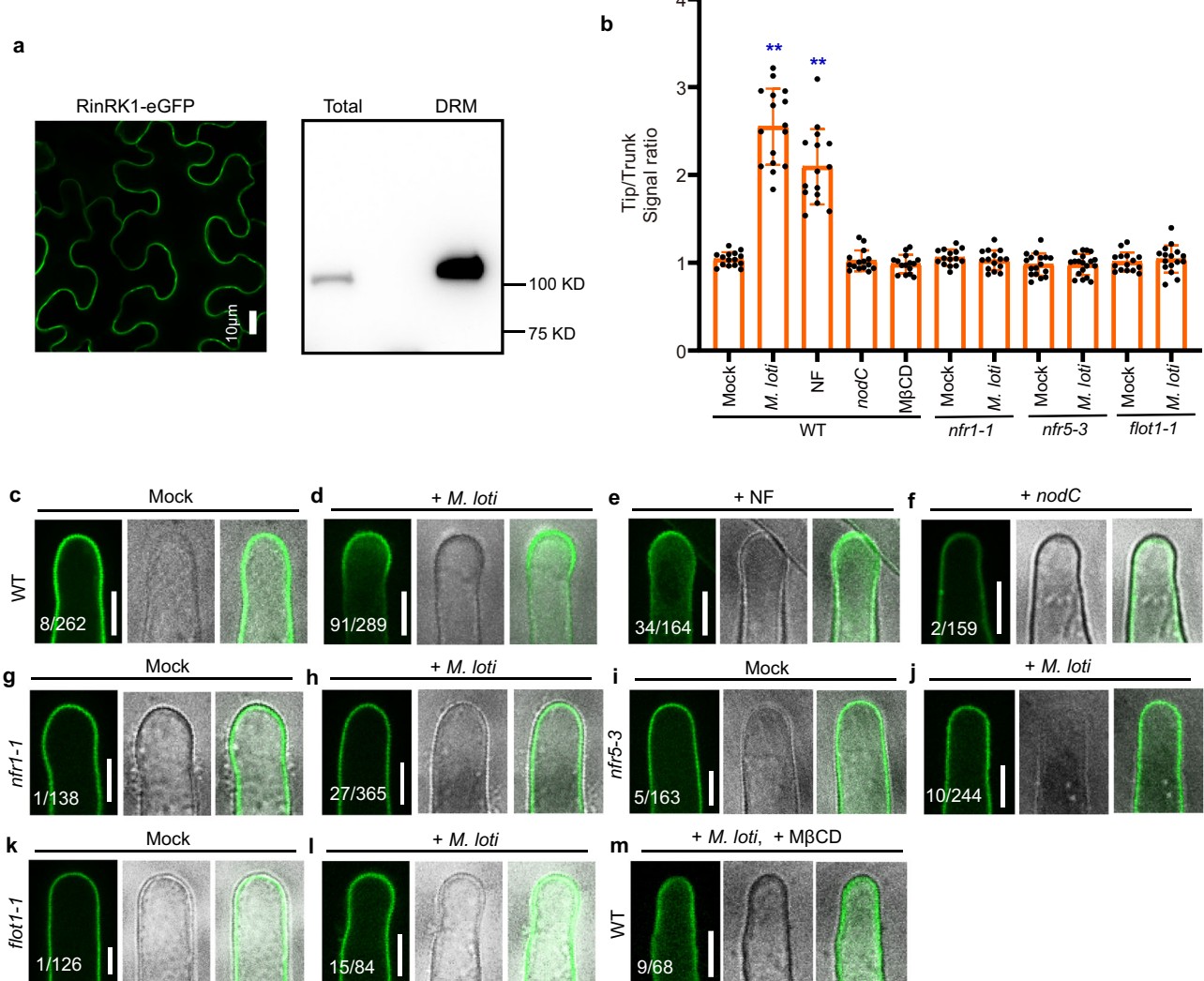

**Fig. 2 | RinRK1 is detected in detergent-resistant membranes and accumulates in root-hair tips in response to *M. loti* and *M. loti* NFs. a** Confocal images of RinRK1-eGFP expressed in *N. benthamiana* leaf. Subcellular localization of RinRK1-eGFP in *N. benthamiana*, showing plasma membrane localization of RinRK1. Immunoblots of RinRK1 protein levels in *N. benthamiana*, with total and detergent-resistant membrane (DRM) proteins analyzed using anti-GFP antibody. **b–m** Assays of *M. loti* NF-induced accumulation of RinRK1 at root-hair tips in *L. japonicus* wild type (WT, Gifu) (**c–f**, **m**), *nfr1-1* (**g**, **h**), *nfr5-3* (**i**, **j**) or *flot1-1* (**k**, **l**). **b** Fluorescence signal ratio of the cell membrane signal density in root-hair tips to the trunks. *n* = 15, 16, 15, 15, 15, 15, 15, 17, 19, 15, 16 biologically independent root hairs examined over two independent experiments. Error bars represent mean ± sd. Asterisks indicate significant differences (**P < 0.01; one-way ANOVA for the comparison between the mock and experimental group). **c–f** Confocal images of RinRK1-eGFP expressed in WT root hairs before (**c**) and after inoculation with *M. loti* R7A (**d**) or *nodC* (**f**), or addition of NF (**e**). RinRK1-eGFP fluorescence was detected in the PM of *L. japonicus* root hairs (**c**). After inoculation with *M. loti* or the addition of NFs, the fluorescence intensity of RinRK1-eGFP increased in the root-hair tips (**d**, **e**) but not after inoculation with *nodC* (**f**). (**g–l**) RinRK1-eGFP in *nfr1-1* (**g**, **h**), *nfr5-3* (**i**, **j**) and *flot1-1* (**k**, **l**) root hairs did not show relocalization to root-hair tips before (**g**, **i**, **k**) or after (**h**, **j**, **l**) *M. loti* inoculation. **m** RinRK1-eGFP expressed in WT *L. japonicus* root hairs treated with *M. loti* and MβCD. Bars = 10 μm. Numbers indicate frequencies of root-hair tip accumulation in all observations. Data were collected based on two biological replicates, with similar frequencies observed in each replicate.

exhibited an even distribution in the PM of root-hair cells (Fig. 2c), consistent with previous findings[32]. However, upon inoculation with *M. loti*, eGFP fluorescence intensity increased at some root-hair tips (Fig. 2d). Occasionally RinRK1 showed a punctate distribution in WT hairy roots, resembling that reported for nanodomain proteins (Supplementary Fig. 3c). This, plus the observed RinRK1 in the detergent-insoluble PM fraction, suggest it is likely present within PM nanodomain-like structures. To validate this hypothesis, we utilized methyl-β-cyclodextrin (MβCD), a compound known for its ability to deplete sterols and disrupt nanodomains[40]. The combination of *M. loti* and MβCD resulted in a significant reduction in RinRK1 accumulation at the root-hair tip, where it is typically induced by *M. loti* (Fig. 2m). These findings strongly suggest that the disruption of sterols in the PM caused by MβCD can diminish the *M. loti*-induced

redistribution of RinRK1 to the tips of root hairs, supporting the hypothesis that RinRK1 accumulates within the PM nanodomain-like structures at the tips of root hairs.

To determine if the redistribution of RinRK1-eGFP in certain root-hair tips is dependent on bacterial NFs, we applied purified NFs from *M. loti*, or utilized a bacterial *M. loti nodC* mutant, which is unable to produce NFs[41]. The results showed that the addition of NFs also induced RinRK1-eGFP accumulation in the tips of root hairs (Fig. 2e). However, the redistribution of the eGFP signal was barely observed in root hairs when inoculated with the *nodC* mutant (Fig. 2f). To further investigate whether this redistribution relies on NF receptors, we introduced the RinRK1-eGFP construct into NF receptor mutant plants (*nfr1-1* or *nfr5-3*). In the absence of rhizobia inoculation, there was no difference in the subcellular localization of

RinRK1 between wild type and the *nfr1* or *nfr5* mutants (Fig. 2g, i). However, upon rhizobia inoculation, the RinRK1 redistribution to root-hair tips was significantly reduced in the NF receptor mutants (Fig. 2h, j), confirming that this redistribution is dependent on NF signaling. The fluorescence intensity ratios at root-hair tips and trunks confirmed that RinRK1 was present at higher levels at root-hair tips following inoculation with *M. loti* or addition of its NFs, but not when using a rhizobia *nodC* mutant, *nfr1*, *nfr5*, or with MβCD treatment (Fig. 2b). These observations provide evidence that the NF signaling is essential and sufficient for promoting RinRK1 accumulation at root-hair tips.

## NFs promote the accumulation of NFR1 and NFR5 at the root-hair tips

The appropriate localization of RLKs in PM nanodomains is important for signal transduction pathways[42,43]. We examined whether the NF receptors NFR1 and NFR5 accumulate at root-hair tips in response to *M. loti* by expressing NFR1-eGFP and NFR5-eGFP under the control of the 35S promoter in wild-type *L. japonicus* roots. The constructs were validated by the complementation of nodulation and infection of the *nfr1-1* and *nfr5-3* nodulation-defective mutants (Supplementary Fig. 4). In the absence of *M. loti*, the fluorescence of NFR1-eGFP and NFR5-eGFP was weak and was evenly distributed in the PM of root hairs (Fig. 3a, k). However, upon inoculation with *M. loti*, a strong fluorescence signal of both NFR1-eGFP and NFR5-eGFP was observed at the tips of root hairs (Fig. 3b, l). Similar results were consistently observed when purified NFs were added (Fig. 3c, m), and occasionally we observed a punctate distribution of NFR1 or NFR5 at root-hair tips (Supplementary Fig. 5a, b).

We further conducted a BiFC assay in *L. japonicus* roots to examine the effects of *M. loti* and purified NFs on the interaction between RinRK1 and NFR1 or NFR5. We created C-terminal split-Venus fragments fused to RinRK1 and N-terminal split-Venus fragments fused to NFR1 or NFR5. Before rhizobia inoculation, Venus fluorescence was evenly distributed in the PM of root-hair cells (Fig. 4a, g). At 48 h post-inoculation with *M. loti* R7A or 24 h after the addition of NFs, strong Venus fluorescence was detected at elongating root-hair tips (Fig. 4b, c, h, i), and a punctate distribution was occasionally observed for NFR1-RinRK1 or NFR5-RinRK1 at root-hair tips (Supplementary Fig. 5c, d). The increased fluorescence intensity ratio showed that both *M. loti* and NFs can promote the accumulation of NFR1-RinRK1 and of NFR5-RinRK1 at the tips of root hairs (Fig. 4f, l). These observations confirm that *M. loti* NFs are essential and sufficient to promote the interactions of NFR1-RinRK1 and NFR5-RinRK1 at the root-hair tips.

## Flot1 is required for IT formation

Flotillins are often used as PM nanodomain or microdomain markers[17,44]. We investigated whether an interaction with flotillin proteins contributed to the *M. loti*-induced relocalization of RinRK1, NFR1, and NFR5 to the tips of root hairs. Using BLAST analysis, we identified three flotillin genes in the *L. japonicus* genome (Fig. 5a). Among these genes, *Flot1* (LotjaGi1g1v0380900), was found to be induced in root hairs by *M. loti* based on gene expression data from Lotus Base (https://lotus.au.dk/). *Flot1* was also highly expressed in nodule primordia and mature nodules (Supplementary Fig. 6a, b). The other two flotillin genes, LotjaGi1g1v0381000 was primarily expressed in immature flowers, and LotjaGi1g1v0381100 was expressed in root hairs but was not induced by *M. loti* (Supplementary Fig. 6a, b). Quantitative real-time polymerase chain reaction (qRT-PCR) analysis confirmed the *Flot1* expression increased in *L. japonicus* roots after inoculation with *M. loti* (Fig. 5b and Supplementary Fig. 6c, d). Thus, *Flot1* was chosen for further investigations.

To assess the role of Flot1 in *M. loti* infections, we first examined the *Flot1* expression pattern by introducing a *Flot1* promoter-GUS fusion construct into *L. japonicus* roots. In uninoculated roots, GUS activity was extremely low, as indicated by histochemical staining (Fig. 5c). However, after inoculation with *M. loti*, GUS activity was clearly detectable in root hairs at 3 days post-inoculation (dpi), in nodule primordia at 5 dpi, and in the vascular bundles of mature nodules at 14 dpi (Fig. 5d-g).

To further investigate the role of Flot1 in rhizobia infection, we obtained a *Flot1* mutant called *flot1-1*, which was generated through a *LORE1* insertion (30129215) in the conserved SPFH/PHB domain (Supplementary Fig. 7a). The expression of *Flot1* was reduced in the *flot1-1* mutant (Supplementary Fig. 7b). After inoculation with *M. loti/LacZ*, the *flot1-1* mutant exhibited larger infection foci (IF) but had significantly fewer IF and ITs compared to the wild-type control (Fig. 5h, i). This reduction of infection events was rescued when the *flot1-1* roots were transformed with *Flot1* (Supplementary Fig. 7d, e). To confirm that Flot1 affects *M. loti* infection, we utilized a CRISPR/Cas9-based knock-out construct to disrupt *Flot1* in wild-type *L. japonicus*. The homozygous mutation at *Flot1* was determined by sequencing (Supplementary Fig. 7c). The *Flot1* knock-out roots displayed a similar reduction in infection events and larger IF, as observed in the *flot1-1* mutant (Fig. 5j). Overall, we conclude that *Flot1* expression in *L. japonicus* is induced by *M. loti* and is necessary for the IT formation.

## Flot1 accumulates at the root-hair tips in response to NFs

To determine whether Flot1 present in PM nanodomains, we expressed Flot1-eGFP in *N. benthamiana* leaves, using a construct previously demonstrated to rescue the *flot1-1* infection phenotype (Supplementary Fig. 7). The results showed that Flot1-eGFP localized to the PM (Supplementary Fig. 8a). Furthermore, Flot1-eGFP was immunologically detected in the detergent-insoluble PM fraction (Supplementary Fig. 8b), indicating its association with membrane nanodomains. Although expression of RinRK1-eGFP or Flot1-eGFP alone resulted in a uniform distribution in the PM, co-expression of RinRK1-eGFP and Flot1-mCherry in *N. benthamiana* leaves revealed a punctate co-localization in the PM (Supplementary Fig. 8c, d). This result indicates a potential association between RinRK1 and Flot1 within membrane nanodomains.

The localization of Flot1 was analyzed using Flot1-eGFP expressed in transformed roots of *L. japonicus*. Flot1-eGFP fluorescence was evenly distributed in the PM prior to inoculation (Fig. 6a). After inoculation with *M. loti*, Flot1-eGFP fluorescence increased at the root-hair tip (Fig. 6b), indicating that, similar to NF receptors and RinRK1, *M. loti* induces the accumulation of Flot1 at the tips of root hairs. Moreover, when *M. loti* and MβCD were combined, the accumulation of Flot1 induced by *M. loti* at the tips of root hairs was significantly reduced (Fig. 6m), indicating that *M. loti* induced the accumulation of Flot1 at root-hair tip within PM nanodomains. To determine whether this relocalization depends on NFs, we added purified NFs to the roots or inoculated them with *M. loti nodC*. The purified NFs promoted accumulation of Flot1-eGFP fluorescence at the tips of root hairs (Fig. 6d), but no such change was observed following the inoculation with the *nodC* mutant (Fig. 6c), implying that NFs are necessary and sufficient to induce the observed accumulation of Flot1 at the root-hair tips.

To test if NF receptors and/or RinRK1 are required for the accumulation of Flot1 at root-hair tips, we expressed Flot1-eGFP in *nfr1-1*, *nfr5-3*, and *rinrk1-1* roots and examined Flot1 localization using confocal imaging. The *nfr1-1*, *nfr5-3* and *rinrk1-1* mutations blocked both *M. loti* and NF-induced accumulation of Flot1-eGFP at the root-hair tips (Fig. 6e, f, h, i, k, l). Consistent with these results, no accumulation was seen in *rinrk1-1* and *nfr1-1* upon inoculation with *nodC* (Fig. 6g, j). The increased fluorescence intensity ratio at the root-hair tips and trunks confirmed our observations (Fig. 6n). These observations suggest that both *M. loti* and NFs can promote Flot1 accumulation at root-hair tips in a process that requires both NF receptors and RinRK1.

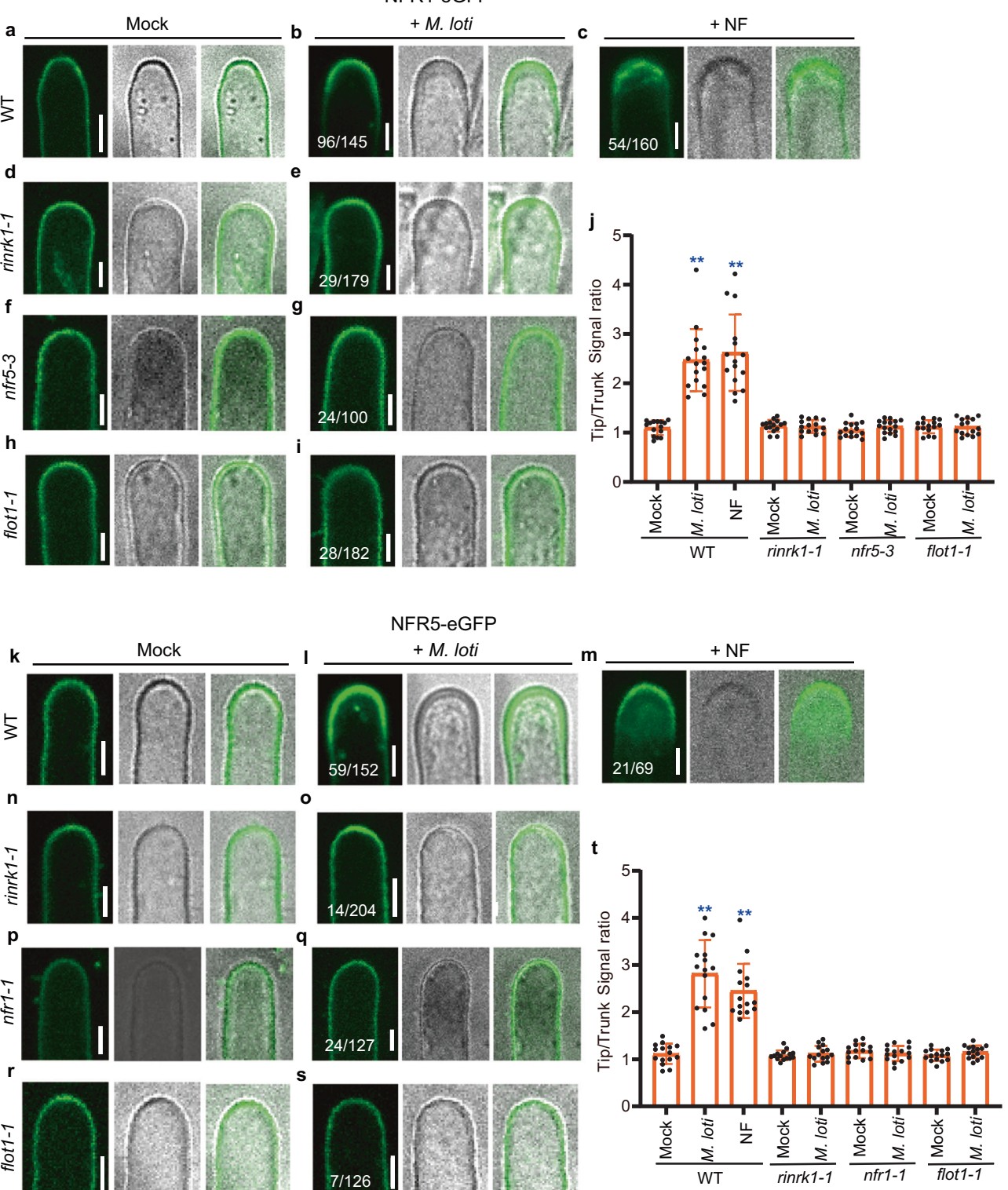

## RinRK1 and Flot1 are required for *M. loti*-induced accumulation of NFR1 and NFR5 at root-hair tips

To investigate the role of RinRK1 in the enhanced accumulation of NFR1 and NFR5 at root-hair tips induced by *M. loti* NF, we expressed NFR1-eGFP and NFR5-eGFP in *rinrk1-1*, and *nfr5-3 or nfr1-1* roots. In *rinrk1-1* and *nfr* mutants, the accumulation of NFR1 and NFR5 at the root-hair tip induced by *M. loti* was lower compared to the wild type (Fig. 3d–g, n–q). This observation was further supported by the analysis of the fluorescence signal ratios of root-hair tips and trunks

(Fig. 3j, t). This finding suggests that the redistribution of NFR1 and NFR5 to the root-hair tip induced by *M. loti* is dependent on RinRK1.

We then conducted an analysis to determine whether Flot1 is essential for *M. loti*-induced relocalization of RinRK1, NFR1, and NFR5, as well as the interactions between RinRK1 and NFR1/NFR5 at the root-hair tip. The results revealed a reduction in the accumulation of *M. loti*-induced RinRK1, NFR1, and NFR5 in *flot1-1* mutant (Figs. 2k, l and 3h, i, r, s). Moreover, there was a significant decrease in *M. loti*-induced accumulation of NFR1-RinRK1 and NFR5-RinRK1 at the root-hair tip in

**Fig. 3 | *M. loti*- and *M. loti* NFs-induce accumulation of NFR1 and NFR5 at root-hair tips in a RinRK1- and Flot1-dependent manner. a–c, k–m** Live-cell confocal images of NFR1-eGFP (**a–c**) or NFR5-eGFP (**k–m**) expressed in WT root hairs before (**a**, **k**) and after inoculation with *M. loti* R7A (**b**, **l**) or addition of NFs (**c**, **m**). NFR1-eGFP (**a**) or NFR5-eGFP (**k**) showed an even distribution in the root-hair PM before inoculation with *M. loti*. After inoculation with *M. loti* or the addition of NFs, the fluorescence intensity of NFR1-eGFP (**b**, **c**) or NFR5-eGFP (**l**, **m**) increased at the root-hair tips. **d–i, n–s** NFR1-eGFP (**d–i**) or NFR5-eGFP (**n–s**) expressed in *rinrk1-1*, *nfr5-3*, *nfr1-1* or *flot1-1* root hairs before (**d**, **f**, **h**, **n**, **p**, **r**) and after inoculation with *M. loti* R7A (**e**, **g**, **i**, **o**, **q**, **s**). There were no significant changes in the accumulation of NFR1-eGFP or NFR5-eGFP fluorescence at the root-hair tips in *rinrk1-1*, *nfr5-3*, *nfr1-1* or *flot1-1* roots after inoculation with *M. loti*. **j, t** The fluorescence ratios of NFR1-eGFP (**j**) or NFR5-eGFP (**t**) were determined by comparing the cell membrane signal density in root-hair tips with the trunks before and after inoculation with *M. loti* R7A or the addition of purified NFs. **j** *n* = 15, 16, 15, 15, 15, 15, 15, 15, 16 biologically independent root hairs examined over two independent experiments. **t** *n* = 15, 15, 15, 15, 15, 15, 15, 15, 15 biologically independent root hairs examined over two independent experiments. Error bars represent mean ± sd. Asterisks indicate significant differences (\*\**P* < 0.01; one-way ANOVA for the comparison between the mock and experimental group). Bars = 5 μm. Numbers indicate frequencies of root-hair tip accumulation in all observations. Data were collected from three biological replicates, with similar frequencies observed in each replicate.

*flot1-1* compared to wild-type roots (Fig. 4e, k). The reduced accumulation observed in *flot1-1* roots was further confirmed by fluorescence intensity ratios (Figs. 2b, 3j, t, and 4f, l). We conclude that Flot1 is crucial for the redistribution of NFR1, NFR5, and RinRK1 to the root-hair tip induced by *M. loti*. Additionally, *M. loti* NFs enhance the accumulation of NFR1-RinRK1 and NFR5-RinRK1 at the root-hair tip in a Flot1-dependent manner.

### Flot1 physically interacts with NFR1, NFR5 and RinRK1, and RinRK1 promotes these interactions

Based on the dependency of the RinRK1-NFR1/NFR5 interaction on Flot1, we hypothesized that Flot1 may physically interact with NFR1, NFR5, and/or RinRK1. We first assayed the potential interactions between the intracellular kinase domains of NFR1, NFR5, and RinRK1 with Flot1 using the GAL4 Y2H system. The results demonstrated interactions between the intracellular domains of RinRK1, NFR1, NFR5 and Flot1 (Fig. 7a). To confirm these interactions, we conducted Co-IP and BiFC assays using full-length proteins in *N. benthamiana*. Co-expression of Flot1-eGFP with NFR1-Myc, NFR5-Myc, or RinRK1-Myc resulted in the co-immunoprecipitation of these three symbiotic RLKs with Flot1 (Fig. 7b). No interaction was observed between Flot1 and MtLYK10 in any of the Y2H, Co-IP or BiFC assays (Fig. 7b and Supplementary Fig. 9). Notably, the presence of RinRK1 enhanced the interaction between NFR1 or NFR5 and Flot1 (Fig. 7c), suggesting that RinRK1 facilitates the interactions of NFR1 and NFR5 with Flot1.

We further evaluated the interactions between Flot1 and NFR1, NFR5, and RinRK1 in *L. japonicus* roots using BiFC assay. These constructs were expressed in wild-type *L. japonicus* through hairy root transformation. Prior to inoculation, we observed Venus fluorescence indicating interactions between NFR1, NFR5, or RinRK1, and Flot1 throughout the root-hair PM (Fig. 7d, i, n). Following inoculation with *M. loti*, the Venus fluorescence resulting from NFR1-Flot1, NFR5-Flot1, and Flot1-RinRK1 interactions accumulated at the root-hair tip (Fig. 7e, j, o), which was confirmed by fluorescence intensity ratio analysis (Fig. 7h, m, p).

To determine whether RinRK1 is required for the *M. loti*-induced NFR1-Flot1 and NFR5-Flot1 interactions at the root-hair tip, we utilized the same BiFC constructs in the *rinrk1-1* mutant. Prior to *M. loti* inoculation, NFR1-Flot1 and NFR5-Flot1 interactions were observed throughout the PM in the *rinrk1-1* mutant (Fig. 7f, k). After inoculation with *M. loti*, the accumulation of NFR1-Flot1 and NFR5-Flot1 at the root-hair tip was diminished in the *rinrk1-1* compared to the wild type (Fig. 7g, l). This observation was supported by fluorescence intensity ratio analysis, which validated that *M. loti* promoted the interaction of Flot1 with NFR1 and NFR5 at the root-hair tip in the wild type, whereas this interaction was significantly reduced in the *rinrk1-1* mutant (Fig. 7h, m). Therefore, we conclude that RinRK1 is necessary for the *M. loti*-induced accumulation of NFR1-Flot1 and NFR5-Flot1 at the tips of root hairs.

### NFR1 Ser/Thr phosphorylation activity is required for *M. loti*-induced NFR1 accumulation at root-hair tips

Previous studies have shown that NFR1 possesses serine/threonine (Ser/Thr) phosphorylation activity, unlike NFR5 and RinRK1[32,45]. In

addition, it has been observed in mammals that Fyn kinase can directly phosphorylate key tyrosine (Tyr) residues in Flot1 and Flot2, impacting their subcellular localization and function[46]. Drawing from these findings, we aimed to investigate whether phosphorylation is necessary for the *M. loti*-induced accumulation of NFR1 and Flot1 at the tips of root hairs.

First, we examined the Tyr phosphorylation activity of NFR1 by expressing and purifying the NFR1 cytoplasmic domain in *Escherichia coli*. We utilized anti-phosphotyrosine (pTyr) and anti-phosphoserine/threonine (pSer/Thr) antibodies to assay the Tyr and Ser/Thr phosphorylation status of NFR1 in vitro. The results demonstrated that NFR1 exhibits autophosphorylation capacity for both Tyr and Ser/Thr residue(s) (Fig. 8a), indicating that NFR1 possesses dual-kinase activity. We then aimed to identify potential Tyr phosphorylation sites in NFR1 by referring to a study conducted on AtCERK1, which identified the essential role of Y428 in its function[47]. By aligning the protein sequences of NFR1 and AtCERK1, we determined that NFR1-Y429 corresponds to AtCERK1-Y428 (Supplementary Fig. 10). Consequently, we generated a variant of NFR1, NFR1$^{Y429F}$, in which this Tyr residue was replaced with phenylalanine (F). As a negative control, we also created NFR1$^{T481A}$, in which the essential Thr 481 residue for Ser/Thr phosphorylation activity was replaced with alanine (A)[45]. In vitro phosphorylation assays revealed that NFR1$^{Y429F}$-CD and NFR1$^{T481A}$-CD lost their Ser/Thr phosphorylation ability, while their Tyr phosphorylation capacity was diminished (Fig. 8a).

To investigate whether NFR1 can phosphorylate Flot1 in vitro, we expressed and purified the Flot1 protein from *E. coli*. However, our results did not provide evidence of NFR1-mediated phosphorylation of Tyr or Ser/Thr residues in Flot1, even in the presence of RinRK1 or NFR5 (Fig. 8b).

To further explore the impact of specific NFR1 mutations, we investigated their effects on the accumulation of NFR1 at the root-hair tip in response to *M. loti*. We introduced NFR1$^{Y429F}$-eGFP and NFR1$^{T481A}$-eGFP constructs into wild-type *L. japonicus* roots and then inoculated the composite plants with *M. loti*. The results revealed a significant reduction in the accumulation of NFR1$^{T481A}$-eGFP at the root-hair tips compared to NFR1-eGFP (Fig. 8c, d, g, h, i). In contrast, NFR1$^{Y429F}$-eGFP exhibited similar levels of accumulation as NFR1-eGFP at the tips of root hairs (Fig. 8e, f, i). Moreover, the expression of NFR1$^{T481A}$-eGFP or NFR1$^{Y429F}$-eGFP in the *nfr1-1* mutants failed to rescue their defective symbiosis (Supplementary Fig. 11). These results suggest that while the Tyr 429 site in NFR1 is indispensable for its biochemical and symbiotic functions, it is not necessary for the *M. loti*-induced accumulation of NFR1 at root-hair tip. However, these findings indicate that NFR1's Ser/Thr phosphorylation activity is essential for the *M. loti*-induced accumulation at the root-hair tip and for symbiotic interactions with *M. loti*.

## Discussion

Rhizobial invasion of legume root hairs requires the precise regulation of molecular and morphological changes to create an optimal environment for the initial bacterial entry and for maintaining fast-growing ITs. In our study, we identified an interaction between RinRK1 and NFR1/NFR5, indicating the formation of a protein

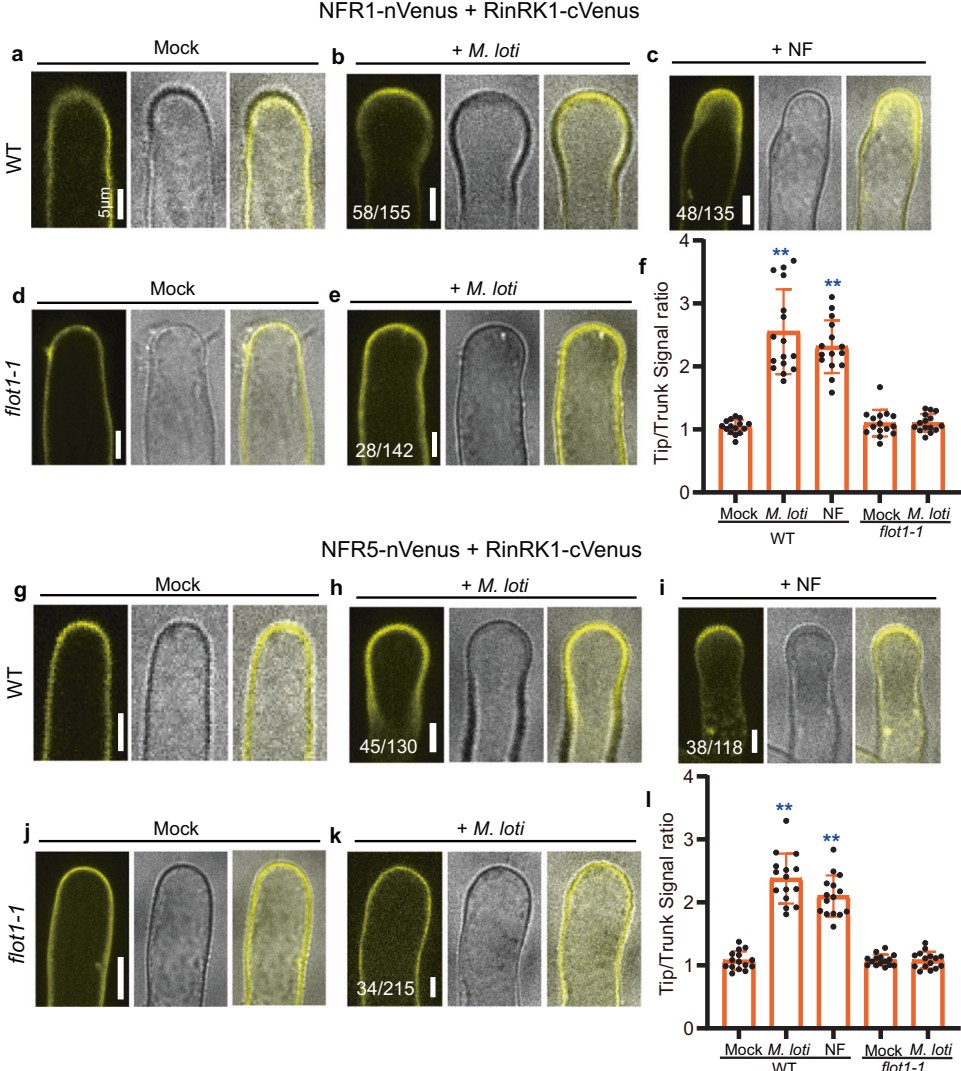

**Fig. 4 | *M. loti* and *M. loti* NFs promote the interaction between RinRK1 and NFR1/NFR5 at root-hair tips in a Flot1-dependent manner. a–e, g–k** BiFC assay using RinRK1-cVenus and NFR1-nVenus or NFR5-nVenus in *A. rhizogenes* transformed roots of WT or *flot1-1*. NFR1-RinRK1 or NFR5-RinRK1 Venus fluorescence was evenly distributed throughout the PM of WT (**a, g**) or *flot1-1* (**d, j**) root hairs before inoculation. After inoculation with *M. loti* R7A (**b, e, h, k**) or the addition of NF (**c, i**), the fluorescence intensity increased in WT (**b, c, h, i**), but not in *flot1-1* (**e, k**). **f, l** The fluorescence ratio of NFR1-RinRK1 (**f**) or NFR5-RinRK1 (**l**) was determined by comparing the cell membrane signal density in root-hair tips with the trunks before and after inoculation with *M. loti* R7A or addition of purified NFs. **f** *n* = 15, 16, 15, 15, 15 biologically independent root hairs examined over two independent experiments. **l** *n* = 15, 15, 15, 15, 15 biologically independent root hairs examined over two independent experiments. Error bars represent mean ± sd. Asterisks indicate significant differences (**$P$ < 0.01; one-way ANOVA for the comparison between the mock and experimental group). Bars = 5 μm. Numbers indicate frequencies of root-hair tip accumulation in all observations. Data were collected from three biological replicates, with similar frequencies observed in each replicate.

complex. This finding strongly suggests that RinRK1 plays a significant role in the pathway that facilitates rhizobial infections via ITs. These results align with previous research[33] that highlights RinRK1 as a key player in IT-mediated infections, as opposed to crack entry. This regulation of RinRK1, in conjunction with its interaction with NFR1 and NFR5, likely contributes to the establishment and maintenance of a favorable environment that supports successful rhizobial colonization and IT formation in legume root hairs.

Flot1, a protein marker for nanodomains, is capable of interacting with NFR1, NFR5, and RinRK1. Furthermore, we observed that upon rhizobial inoculation, NFR1 and NFR5 accumulated at the root-hair tip, and this accumulation was dependent on RinRK1 and Flot1. Our findings differed from a previous study conducted in *M. truncatula*, where NFs alone were insufficient to induce the movement of LYK3 toward the root-hair tip, and Flot4 puncta were affected only in a specific mutant, *hcl-1*, which carries a kinase-dead mutation in *LYK3*.

In other *LYK3* alleles and the *nfp* mutant, Flot4 puncta were unaffected[22]. However, in our study, the application of purified NFs triggered the accumulation of NFR1, NFR5, RinRK1, and Flot1 alone or as interactors at the tips of root hairs. Consistent with this, we observed a substantial decrease in the accumulation of RinRK1 or Flot1 at root-hair tips in *nfr1-1* and *nfr5-3* mutants or after inoculation with the *M. loti nodC* mutant. The discrepancies between our findings and previous work may be related to differences in NF sensitivity between *Lotus* and *Medicago* species, as NFs alone can induce nodule formation in *L. japonicus* and *M. sativa*, but not in *M. truncatula*[48,49]. Another possible reason for the discrepancies is that we are working with a different flotillin protein. Phylogenetic analysis indicates that LjFlot1 is more closely related to MtFlot5 than to MtFlot2 or MtFlot4. This is supported by the distinct infection phenotype observed in flotillin mutants. *Ljflot1* shows few infection events but larger infection foci, while *Mtflot4* exhibits larger and more infection foci but

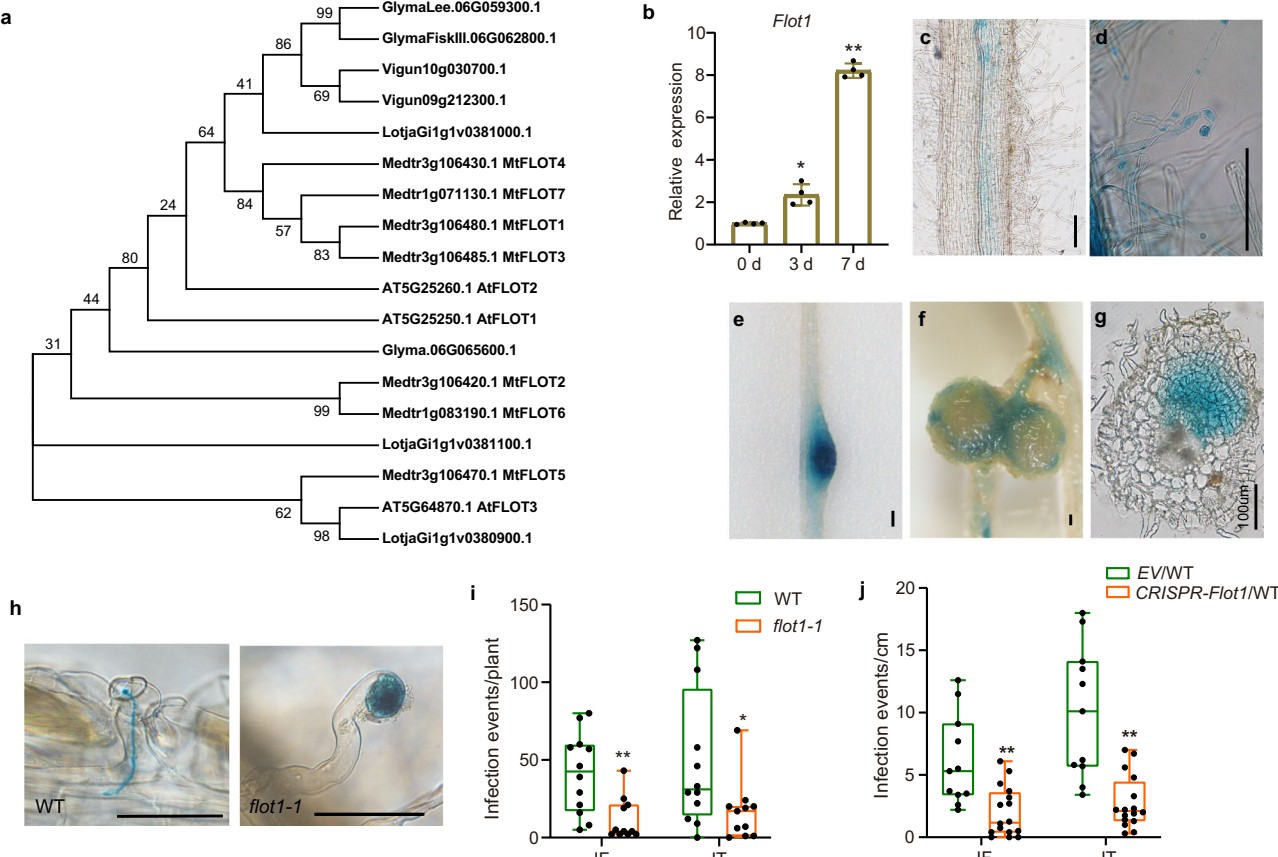

**Fig. 5 | Flot1 is required for rhizobia infection. a** Phylogenetic analysis of flotillin proteins from legume species *L. japonicus*, *M. truncatula*, *G. max*, and *Vigna unguiculata*, and the non-legume species *A. thaliana*. **b** qRT-PCR analysis of *Flot1* transcript levels in the roots of wild-type *L. japonicus* after inoculation with *M. loti* R7A (0, 3 and 7 days). n = 4 from two biological replicates. Expression levels were normalized against the reference gene *Ubiquitin*. Error bars represents mean ± sd. **c**–**g** p*Flot1:GUS* expression pattern in wild-type roots before (**c**) and after (**d**–**g**) inoculation with *M. loti* R7A. GUS activity was almost undetectable in uninoculated *L. japonicus* roots (**c**). GUS activity was detected in the root hairs (**d**), nodule primordia (**e**), and the vascular bundles of mature nodules (**f**). A nodule section showing p*Flot1:GUS* expression in all of the cell layers in a nodule primordium (**g**). **h**–**j** Infection thread phenotypes (**h**) and infection events in *flot1-1* (**i**) or CRISPR/

Cas9-*Flot1* in wild-type *L. japonicus* (**j**) at 5 dpi with *M. loti* R7A/*lacZ*. Normal elongating infection threads in the wild type, while enlarged infection foci were seen in *flot1-1* at 5 dpi. **i, j** The *flot1-1* (**i**) and CRISPR/Cas9-based *Flot1* knock-out roots (**j**) produced fewer infection events (e.g., infection foci and infection threads) compared to the WT control. **i** n = 12 (WT), n = 11 (*flot1-1*). **j** n = 11 (*EV*/WT), n = 16 (*CRISPR-Flot1*/WT). Bars = 100 μm (**c**–**g**) or 50 μm (**h**). IF, infection foci; IT, infection thread. **b, i, j** Asterisks indicate significant differences (*$P < 0.05$, **$P < 0.01$; two-tailed Student's *t*-test for the comparison between the *EV* or WT and experimental group). One representative example out of three independent experiments is shown. Boxplots show the median, upper and lower quartiles, and whiskers indicating the maximum and minimum values.

fewer ITs. Further studies are required to elucidate the specific roles of different flotillins in symbiotic infection.

Our study also revealed that RinRK1 and Flot1 are potential nanodomain proteins that are induced by NFs and accumulate at the tips of root hairs. Both RinRK1 and Flot1 were detected in detergent-insoluble fractions. Co-expression of RinRK1 and Flot1 in *N. benthamiana* leaves demonstrated their co-localization in punctate structures. Notably, the accumulation of RinRK1 and Flot1 at root-hair tips induced by *M. loti* was significantly reduced when combined with MβCD, an agent that depletes sterols and disrupts nanodomain. However, it is worth mentioning that we were unable to consistently detect punctate structures at the root-hair tips, potentially due to the limitations in resolution provided by our confocal microscopy system.

Studies on mammals have revealed that flotillins undergo phosphorylation of Tyr residues by Src family kinases upon growth factor stimulation[50,51]. Plants lack canonical receptor Tyr kinases but possess numerous RLKs that exhibit Ser/Thr kinase activities, some of which can also phosphorylate Tyr residues[52,53]. In *A. thaliana*, the NFR1 homolog, AtCERK1, is an RLK known to exhibit dual-specificity kinase activity[47]. In our study, we identified NFR1 as a dual-specificity kinase capable of phosphorylating both Ser/Thr and Tyr residues. Mutations

in the Ser/Thr phosphorylation site of NFR1 disrupted it from accumulating at root-hair tips following inoculation with *M. loti* and abrogated its ability to rescue the symbiotic defects of *nfr1-1*. These findings indicate that phosphorylation of NFR1 is essential for its translocation to the root-hair tip in response to *M. loti* NF signals and for its proper symbiotic functions. Interestingly, although the phosphorylation of NFR1 is essential, the NFR1$^{Y429F}$ mutant is still able to re-localize to the root-hair tip upon *M. loti* inoculation. In vitro assays revealed that NFR1 possessed autophosphorylation activity but did not phosphorylate the Tyr residue of Flot1 directly. It is possible that other unidentified kinases may be responsible for the phosphorylation of Flot1 at Tyr residue. Our in vitro experiments also showed that both NFR1$^{Y429F}$ and NFR1$^{T481A}$ exhibited reduced Ser/Thr kinase activity and low levels of Tyr kinase activity. This suggests the presence of a regulatory cycle between Ser/Thr and Tyr phosphorylation activity, similar to AtCERK1[47].

Taking these findings together, we propose a model (Fig. 9) in which RinRK1 facilitates the assembly of a protein complex involving NFR1, NFR5, and Flot1 within nanodomain-like structures at root-hair tips following the perception of NF signals. In the absence of *M. loti*, NFR1 and NFR5 are randomly distributed in the PM, while RinRK1 and

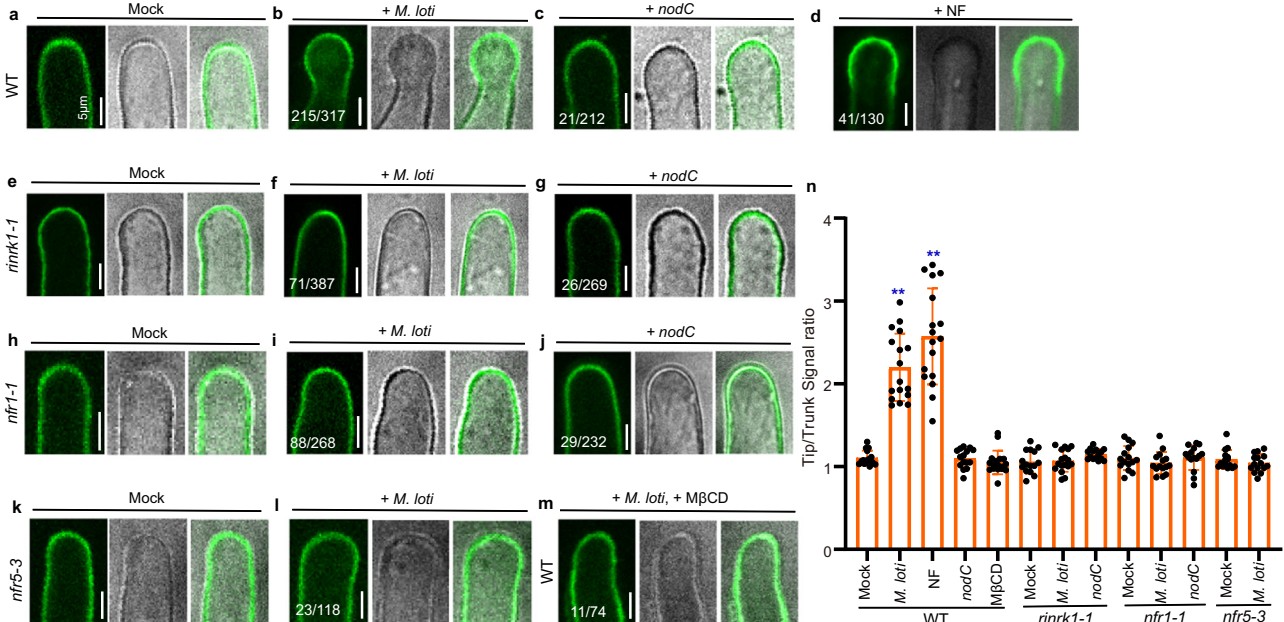

**Fig. 6 | Flot1-eGFP accumulation at root-hair tips in response to *M. loti* or NFs dependent on NFR1, NFR5, or RinRK1. a–d** Flot1-eGFP expressed in WT root hairs before (**a**) and after inoculation with *M. loti* R7A (**b**) or *M. loti nodC* (**c**) or treated with NF (**d**). Flot1-eGFP was evenly distributed in the root-hair PM before inoculation. An increase in Flot1-eGFP fluorescence intensity was observed in the root-hair tips after inoculation with *M. loti* R7A (**b**) or the addition of NFs (**d**), but not after inoculation with *nodC* (**c**). **e–l** Assays of Flot1-eGFP expressed in *rinrk1-1* (**e–g**), *nfr1-1* (**h–j**) or *nfr5-3* (**k–l**) root hairs before (**e, h, k**) and after inoculation with *M. loti* R7A (**f, i, l**) or *M. loti nodC* (**g, j**). The distribution of Flot1-eGFP in the *rinrk1-1*, *nfr1-1*, or *nft5-3* roots remained unchanged after inoculation with *M. loti* R7A (**f, i, l**) or with *M. loti nodC* (**g, j**). **m** Flot1-eGFP expressed in WT *L. japonicus* root hairs treated with *M. loti* and MβCD. Bars = 5 μm. **n** The fluorescence ratios were determined by comparing the cell membrane signal density in root-hair tips with the root-hair trunks before and after inoculation with *M. loti* R7A or the addition of purified NFs, or with *M. loti nodC*. *n* = 15, 17, 17, 15, 17, 15, 17, 15, 15, 15, 15, 15, 15 biologically independent root hairs examined over two independent experiments. Error bars represent mean ± sd. Asterisks indicate significant differences (**$P < 0.01$; one-way ANOVA for the comparison between the mock and the experimental group). Numbers indicate frequencies of root-hair tip accumulation in all observations. Data were collected based on three biological replicates, with similar frequencies found in each replicate.

Flot1 are already present in PM nanodomain-like structures. Upon perception of *M. loti* NFs by NFR1 and NFR5, a symbiosis-specific signaling cascade is activated, leading to the transcriptional upregulation of *RinRK1* and *Flot1*. RinRK1 enables the interaction of Flot1 with NFR1 and NFR5, leading to the recruitment of NF receptors to nanodomain-like structures at the root-hair tip. This enrichment of NF receptor complexes in nanodomains may enhance their activity and bring them closer to the source of NFs (i.e. rhizobia attached to the root hair or within microcolonies). This localization could potentially enhance IT formation, promoting successful symbiotic interactions.

## Methods

### Plant and bacterial materials

*Lotus japonicus* Gifu B-129[54] and the mutant lines *nfr1-1*[8], *nfr5-3*[7], and *rinrk1-1*[32] were used in this study. The *flot1-1* line was obtained from the *L. japonicus* LORE1 transposon insertion pool at Aarhus University, Denmark[55]. *Agrobacterium rhizogenes* strain AR1193[56] was used for the transformation of *L. japonicus* hairy roots, whereas *A. tumefaciens* strain EHA105 was used for the transformation of *N. benthamiana*. *Escherichia coli* DH10B or DH5α cells were used for cloning. *M. loti* strain R7A carrying pXLGD4 (*lacZ*) or mTag was used for inoculations.

### Cloning, DNA manipulation, and plasmid construction

To analyze *Flot1* expression in *L. japonicus* hairy roots, the *Flot1* promoter (2 kb upstream of the start codon) was inserted into the entry vector pDONR207 and then transferred into the pKGWFS7 vector to generate the p*Flot1:GUS* construct via the LR reaction (Invitrogen).

To determine the subcellular localization of NFR1, NFR5, Flot1, and RinRK1 in *L. japonicus* hairy roots, *NFR1*, *NFR5*, and *Flot1* in the pDONR207 vector were inserted into pK7WGF2 to generate the NFR1-

eGFP, NFR5-eGFP, and Flot1-eGFP constructs via the LR reaction. The RinRK1-eGFP construct was generated using the Golden Gate cloning method[57]. The *RinRK1* PCR product was inserted into pL0V-SC3 after *Bbs1* digestion. This vector and the EC16570 vector were digested with *Bsa1* to generate RinRK1-eGFP as a level 1 construct. This construct was incorporated into EC50507 (https://www.ensa.ac.uk/) to add p*35S:NLS-DsRed* as a transgenic marker, thereby generating the level 2 construct p*LjUb:RinRK1-eGFP*. These vectors were also used for the complementation assay involving the transformation of *nfr1-1*, *nfr5-3*, *flot1-1*, and *rinrk1-1* hairy roots.

For co-immunoprecipitation assays in *N. benthamiana*, the PCR products (*NFR1*, *NFR5*, *RinRK1* or *MtLYK10*) were inserted into the destination vector pCambia1305-35S-Myc following *Kpn1* and *Sal1* digestion. The *RinRK1* or *RinRK1-N* PCR products were inserted into the destination vector pUB-GFP/X-Flag, which was modified from pUB-GFP, following *Kpn1* and *Xba1* digestion.

For the luciferase biomolecular complementation assays in *N. benthamiana*, the *RinRK1* PCR product was inserted into luciferase vector pCambia1300-35S-nLUC following *Kpn1* and *Sal1* digestion. The *NFR1*, *NFR5*, and *LYK10* PCR products were inserted into pCambia1300-35S-cLUC following *Kpn1* and *Pst1* digestion.

For the BiFC assays, the *Flot1* PCR product was inserted into pL0V-SC3 after a *Bbs1* digestion. The resulting construct was inserted into EC12849 to generate Flot1-nVenus. The *RinRK1* PCR product was incorporated into pL0V-SC3 after a *Bbs1* digestion, after which the fragment was inserted into EC12850 to generate RinRK1-cVenus. Finally, these fragments were incorporated into EC50507 to add p*35S:DsRed* as a transgenic marker, thereby producing the BiFC construct p*AtUb:Flot1-nVenus*/p*LjUb:RinRK1-cVenus*. Other BiFC vectors were constructed using the same method to generate p*AtUb:NFR1-*

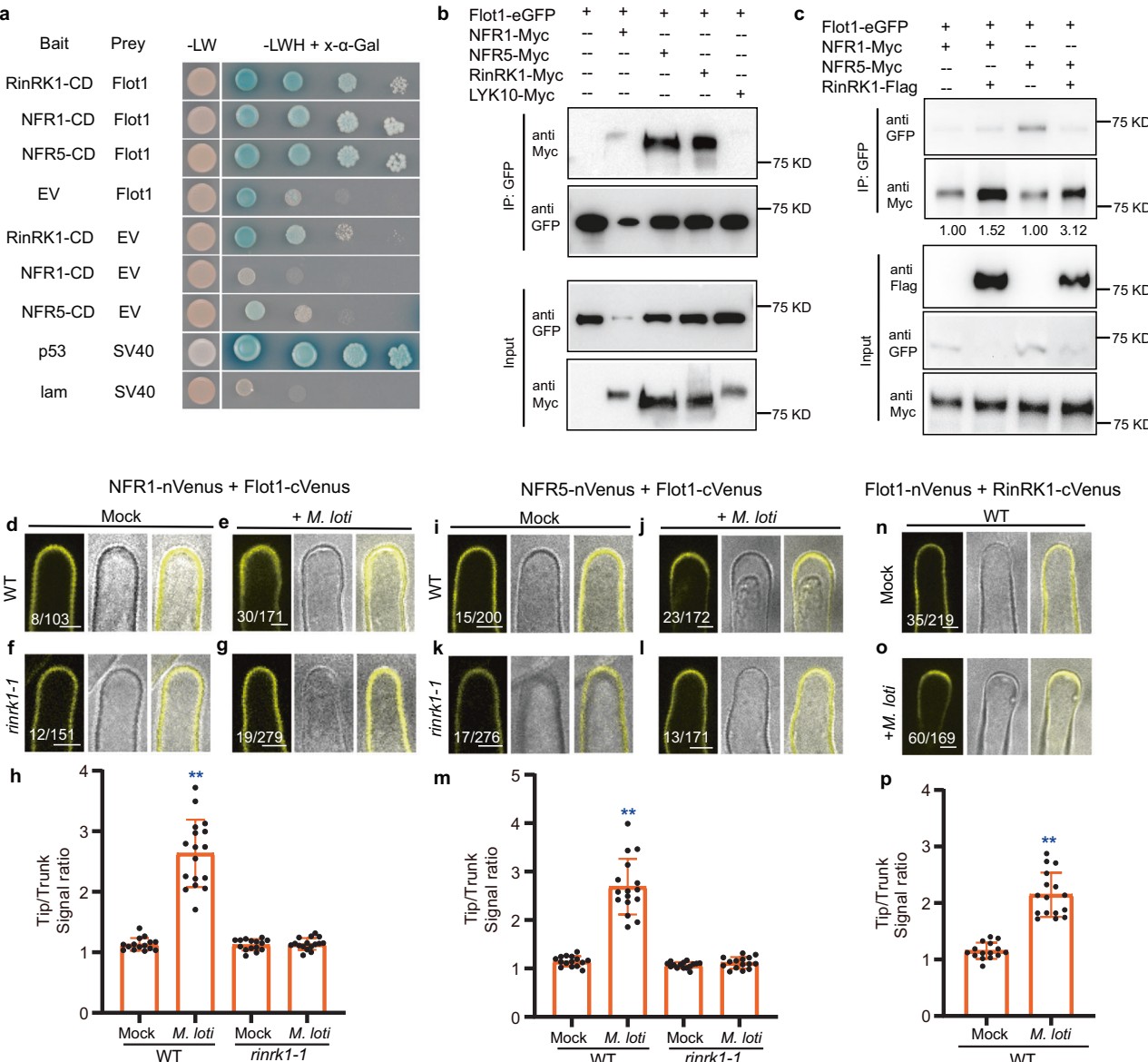

**Fig. 7 | Flot1 interacts with NFR1, NFR5, and RinRK1. a** GAL4 yeast two-hybrid assays between Flot1 and the cytoplasmic domains (CD) of NFR1, NFR5, and RinRK1. Strong interactions were indicated by growth on SD-LWH media and blue staining with X-α-Gal. **b, c** Co-immunoprecipitation analysis of interactions between Flot1-GFP and NFR1-Myc, NFR5-Myc, RinRK1-Myc, and MtLYK10-Myc expressed in *N. benthamiana* leaves (**b**). Co-IP was conducted with anti-GFP agarose using total extracts, and immunoblotting was performed utilizing anti-Myc or anti-GFP antibody. The values presented below the blot correspond to the relative quantities of NFR1-Myc or NFR5-Myc co-immunoprecipitated with Flot1-eGFP in the absence (lanes 1 and 3) or presence (lanes 2 and 4) of RinRK1. These values were normalized by comparing lane 2 to lane 1, and lane 4 to lane 3. Stronger interactions between Flot1 and NFR1 or NFR5 were observed when RinRK1-Flag was present (**c**). **d–o** BiFC assays to detect interactions between Flot1 and NFR1 or NFR5 or RinRK1. NFR1-Flot1, NFR5-Flot1, or Flot1-RinRK1 Venus fluorescence was evenly distributed in the entire

PM of WT (**d, i, n**) or *rinrk1-1* (**f, k**) root hairs before inoculation. After inoculation with *M. loti* R7A, NFR1-Flot1, NFR5-Flot1, or Flot1-RinRK1 Venus fluorescence accumulated at the root-hair tips in the WT (**e, j, o**), but not in *rinrk1-1* (**g, l**). Bars = 5 µm. **h, m, p** The fluorescence ratio in the root-hair tips to the trunks, demonstrating that *M. loti* can promote the accumulation of NFR1-Flot1 (**h**), NFR5-Flot1 (**m**), or Flot1-RinRK1 (**p**) at root-hair tips. **h** *n* = 15, 17, 15, 16 biologically independent root hairs examined over two independent experiments. **m** *n* = 15, 16, 15, 15 biologically independent root hairs examined over two independent experiments. **p** *n* = 15, 16 biologically independent root hairs examined over two independent experiments. Error bars represent mean ± sd. Asterisks indicate significant differences (**$P < 0.01$; two-tailed Student's *t*-test for the comparison between the mock and experimental group). Numbers indicate frequencies of root-hair tip accumulation in all observations. Data were collected based on two biological replicates, with similar frequencies found in each replicate.

*nVenus*/p*LjUb:RinRK1-cVenus*, p*AtUb:NFR5-nVenus*/p*LjUb:RinRK1-cVenus*, p*AtUb:NFR1-nVenus*/p*LjUb:Flot1-cVenus*, and p*AtUb:NFR5-nVenus*/p*LjUb:Flot1-cVenus*.

For CRISPR/Cas9 knock-out construct, the web tool CRISPR-P 2.0 (http://cbi.hzau.edu.cn/crispr/) was used to design primers for the guide RNAs. Two gRNAs were designed to target Flot1. CRISPR/Cas9 vector pKGSD401 and guide RNA-scaffold vector pG-DT1T2 were used for gene KO[58]. Two specific primers were synthesized, gRNA-T1-F and

gRNA-T2-R, and pG-DT1T2 was used as template for amplification of the guide RNA model, which was digested with *BsaI* (NEB) and ligated using T4 DNA ligase (NEB) in one reaction (4 min at 37 °C, 3 min at 16 °C, for 40 cycles) to make the CRISPR target vector pKGSD401-Flot1-T1T2.

All PCR amplifications were completed using high-fidelity DNA polymerase MAX (Vazyme, Nanjing, China) and all constructs were confirmed by DNA sequencing. The constructs in the destination vectors were introduced into *A. rhizogenes* AR1193 cells for the

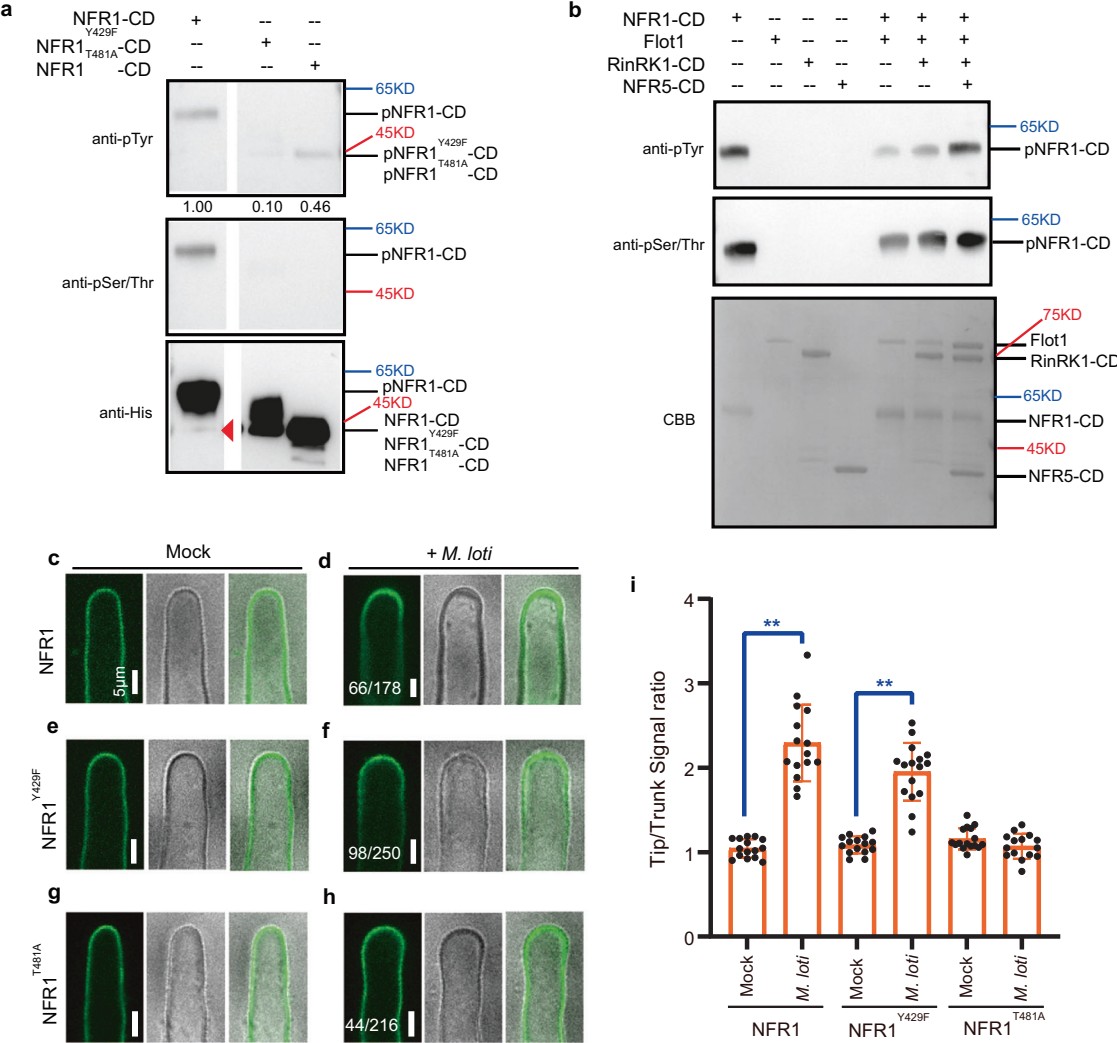

**Fig. 8 | Assays of the role of NFR1 kinase activity in rhizobia-induced NFR1-eGFP accumulation at root-hair tips. a** In vitro kinase assays of the cytoplasmic domain (CD) of NFR1, NFR1$^{Y429F}$, and NFR1$^{T481A}$. Purified His-tagged NFR1, NFR1$^{Y429F}$, or NFR1$^{T481A}$ CDs were incubated with ATP for 30 min. The proteins were detected by immunoblotting with anti-His antibody. Phosphorylated products were detected using anti-pSer/Thr or anti-pTyr antibodies. Un-phosphorylated NFR1-CD band is indicated by the red triangle. **b** Trans-phosphorylation assay of NFR1-CD and Flot1. Purified His-tagged NFR1-CD was incubated with MBP-tagged Flot1 and ATP for 30 min. Phosphorylated products were detected using anti-pSer/Thr or anti-pTyr antibodies. No phosphorylation of Flot1-MBP was observed, even after adding RinRK1 or NFR5 CDs. CBB, Coomassie brilliant blue. Experiments were repeated three times with similar results. **c**–**h** Live-cell confocal images of NFR1, NFR1$^{Y429F}$, or NFR1$^{T481A}$ in wild-type *L. japonicus* hairy roots. NFR1-eGFP (**c**), NFR1$^{Y429F}$-eGFP (**e**), or

NFR1$^{T481A}$-eGFP (**g**) was evenly distributed in the root-hair PM before inoculation. NFR1-eGFP (**d**) or NFR1$^{Y429F}$-eGFP (**f**) fluorescence accumulated at the root-hair tips after inoculation with *M. loti* R7A. However, NFR1$^{T481A}$-eGFP (**h**) fluorescence did not accumulate to root-hair tips after inoculation with *M. loti*. Bars = 5 μm. **i** The fluorescence signal ratios in the root-hair tips to the trunks before and after rhizobial inoculation. *n* = 15, 15, 15, 16, 16, 15 biologically independent root hairs examined over two independent experiments. Error bars represent mean ± sd. Asterisks indicate significant differences (**$P < 0.01$; two-tailed Student's *t*-test for the comparison between the mock and the experimental group). Numbers indicate frequencies of root-hair tip accumulation in all observations. The experiments were performed in three biological replicates, with similar frequencies observed in each replicate.

transformation of *L. japonicus* hairy roots or into *A. tumefaciens* EHA105 cells for the transient expression in *N. benthamiana* by electroporation. All primers are listed in Supplementary Table 1.

## Plant growth, rhizobial inoculation, infection, and nodulation phenotype analysis

*L. japonicus* seeds were scarified with sandpaper, surface-sterilized in 10% (v/v) sodium hypochlorite for 5 min and rinsed five times with sterile water. The sterilized seeds were imbibed in water and then transferred to plates containing 0.8% water agar. The plates were incubated at 4 °C overnight and then at 22 °C in darkness for 3- to 4-days. The germinated seedlings were transferred to a vermiculite and perlite (1:1) mixture. After 1 week, the seedlings were inoculated with *M. loti* R7A carrying pXLGD4 (*lacZ*). The infection phenotypes and events

were determined at the indicated time points by examining the roots stained with 5-bromo-4-chloro-3-indolyl beta-D-galactopyranoside (X-Gal) using a light microscope (ECLIPSE Ni, Nikon, Tokyo, Japan).

To prepare nodule sections, nodules were fixed in 2.5% (v/v) glutaraldehyde and embedded in Technovit 7100 resin (Kulzer GmbH) according to the manufacturer's instructions, after which 5–10 mm transverse sections were prepared using a microtome (Leica RM2265).

## Phylogenetic analyses

The *A. thaliana*, *M. truncatula*, soybean (*G. max*), cowpea (*Vigna unguiculata*), and *L. japonicus* protein sequences were obtained from the TAIR (https://www.arabidopsis.org/index.jsp), Phytozome (v12.1) (https://phytozome-next.jgi.doe.gov/), and Lotus Base (https://lotus.au.dk/) databases. All protein sequences were aligned using ClustalW,

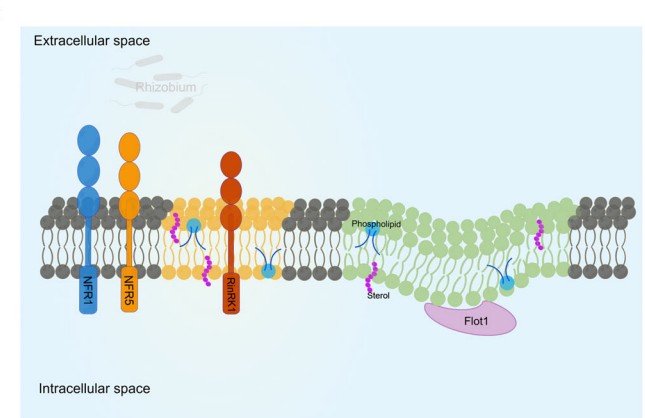

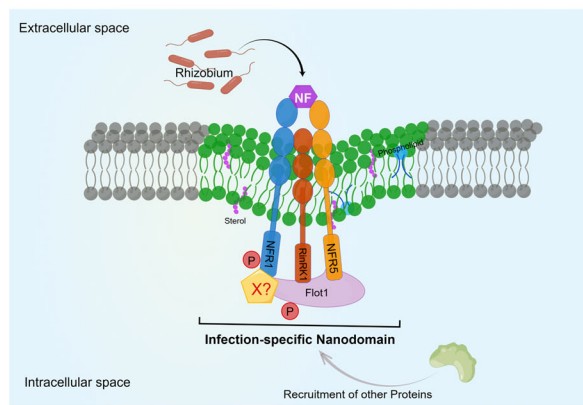

**Fig. 9 | A proposed model of infection-specific nanodomain assembly. a** In the absence of rhizobia, NFR1 and NFR5 are randomly distributed in the plasma membrane, while RinRK1 and Flot1 are present in different nanodomain-like structures. **b** After rhizobial infection, the perception of NFs by NFR1 and NFR5 initiates a symbiosis-specific signaling cascade, leading to the transcriptional upregulation of *RinRK1* and *Flot1*. RinRK1 promotes interaction between NFR1, NFR5, and Flot1, which leads to the recruitment of NF receptors to nanodomain-like structures at the root-hair tips. The formation of infection-specific nanodomains play a role in regulating the production of infection threads. Flot1 may undergo phosphorylation by unknown protein(s). The model was created using Figdraw (www.figdraw.com).

and the phylogenetic analysis was conducted using MEGA7.0. The phylogenetic tree was constructed using the Maximum Likelihood method, with 1000 bootstrap replicates.

### Analysis of promoter:*GUS* and complementation of *L. japonicus* transformed hairy roots

*A. rhizogenes* AR1193 cells carrying the indicated constructs were used to transform *L. japonicus* Gifu, *flot1-1*, *rinrk1-1*, *nfr1-1*, or *nfr5-3* hairy roots on half-strength B5 medium. After 2 weeks, the transformed hairy roots were examined regarding DsRed fluorescence using the LUYOR-3415RG Hand-Held Lamp. The transgenic roots were transferred to a vermiculite and perlite (1:1) mixture and then inoculated with *M. loti* R7A/*lacZ* 5–7 days later. The *L. japonicus* hairy roots transformed with p*Flot1:GUS* were stained with X-Gluc to visualize gene expression at the indicated time points. For the *Flot1* complementation assays, the transgenic roots were stained with X-Gal and the infection events were scored by examining the samples using the Nikon ECLIPSE Ni light microscope at 5 dpi. For the *RinRK1*, *NFR1*, or *NFR5* complementation assays, the number of nodules was determined at 18 days or 3 weeks after the inoculation with *M. loti* R7A/*lacZ*.

### Total RNA extraction and qRT-PCR

Total RNA was extracted using the TRIpure Isolation Reagent (Aidlab, China) according to the manufacturer's instructions and then quantified using the NanoDrop 2000 spectrophotometer (Thermo, Waltham, MA, USA). The first-strand cDNA was synthesized using the TransScript One-step gDNA Removal and cDNA Synthesis SuperMix kit (Trans Gen Biotech). The qRT-PCR analysis was performed using the ABI StepOne Plus PCR system and the SYBR Green Realtime PCR mix (Takara). The *ubiquitin* gene (Lj5g3v2060710) was used as the internal control to normalize the qRT-PCR data for each sample.

### Bimolecular fluorescence complementation assays

The Flot1 and RinRK1, NFR1, and NFR5 BiFC constructs (p*AtUb:Flot1-nVenus*/p*LjUb:RinRK1-cVenus*, p*AtUb:NFR1-nVenus*/p*LjUb:RinRK1-cVenus*, p*AtUb:NFR5-nVenus*/p*LjUb:RinRK1-cVenus*, p*AtUb:NFR1-nVenus*/p*LjUb:Flot1-cVenus*, and p*AtUb:NFR5-nVenus*/p*LjUb:Flot1-cVenus*) were made using Golden Gate cloning. The constructs were introduced into AR1193 by electroporation and then expressed in *L. japonicus* hairy roots. The transgenic hairy roots were analyzed on the basis of the NLS-DsRed marker and inoculated with *M. loti* R7A/mTag (OD$_{600}$ approximately 0.05) or purified *M. loti* R7A NF (10 nM). Samples were analyzed

at 2–3 days after they were inoculated with rhizobia or 24 h after the NF treatment. The excitation/emission filter sets were as follows: Venus (514 nm/524–545 nm), DsRed (561 nm/570–610 nm). The proportion of aggregated proteins at the root-hair tip was analyzed in the infection zone at the same magnification. For each BiFC construct, more than 100 root hairs from six composite plants were analyzed by live-cell imaging each time, and the experiment was repeated at least twice.

### Subcellular localization in *N. benthamiana* leaves and transformed *L. japonicus* hairy roots

To determine the subcellular localization of Flot1 and RinRK1 in *N. benthamiana* leaves, *A. tumefaciens* EHA105 cells were used for the agroinfiltration of *N. benthamiana* leaves with constructs along with p19. After 2–3 days, the leaves were examined using a confocal microscope (TCS SP8, Leica). The excitation/emission filter sets for GFP were 488 nm/498–540 nm. We used an anti-GFP antibody (Abmart, CAT.M20004) for the immunoblot analysis.

To determine the subcellular localization of RinRK1, Flot1, NFR1, and NFR5 in *L. japonicus* hairy roots, *A. rhizogenes* AR1193 cells carrying the indicated constructs were used to transform *L. japonicus* hairy roots. The transgenic roots were then inoculated with *M. loti* R7A/mTag, *M. loti* R7A/NF, or *M. loti* nodC. The subcellular localization of the proteins was determined using a spinning disk confocal microscope (REVOLUTION XD, ANDOR). The excitation/emission filter sets were as follows: GFP (488 nm/498–540 nm), mCherry/DsRed (561 nm/570–610 nm), and mTag (405 nm/415–454 nm). The proportion of aggregated proteins at the root-hair tip was analyzed in the infection zone at the same magnification. The protein subcellular localization assays were repeated at least twice, more than 100 root hairs from six composite plants were analyzed by live-cell imaging each time.

### Yeast two-hybrid assays

The full-length sequences or the sequences encoding the ED and CD of *RinRK1*, *NFR1*, and *NFR5* were amplified by PCR from *L. japonicus* Gifu B-129 and *MtLYK10* was amplified by PCR from *M. truncatula* A17 root cDNA using primers listed in Supplemental Table 1. We used the yeast split-ubiquitin system to analyze protein–protein interactions in yeast strain NMY51 (Weidi Bio, China). The PCR products were digested with *SfiI*, after which *RinRK1* was inserted into the pCCW-STE plasmid to produce the RinRK1-CubG construct, whereas *MtLYK10* was cloned into pDL2xN-STE to generate the MtLYK10-NubG construct. The NFR1-NubG, NFR5-NubG, and CubG-LjSymREM1 constructs were described

previously[24]. Yeast cells were co-transformed with RinRK1-CubG and NFR1-NubG, NFR5-NubG, or MtLYK10-NubG. The transformation and analysis of growth on the selective medium were performed using the DUAL system (protocol P01001-B03). Transformants were examined for potential protein–protein interactions on SD medium (0.67% yeast nitrogen base, 2% glucose, 2% Bacto-agar, and an amino acid mixture) lacking the appropriate auxotrophic markers but containing different 3-AT concentrations.

For the GAL4 system-based yeast two-hybrid assays, the products of the PCR amplification of the ED and CD sequences were cloned into pDONR207 and sequenced to confirm they were correct. They were subsequently inserted into pDEST-GBKT7 or pDEST-GADT7. Yeast two-hybrid assays were performed using yeast strain AH109 (Clontech) following standard procedures (Yeast Protocols Handbook PT3024-1, Clontech). The proteins were extracted by post-alkaline methods[59] and detected by western blot with antibody Myc (Abmart, CAT.M2000M) and HA (HuiOu Biotech, CAT.HOA012HA01).

### Protein extraction and co-immunoprecipitation assay
Different combinations of constructs (NFR1-Myc, NFR5-Myc, MtLYK10-Myc, RinRK1-Flag, RinRK1-N-Flag, and Flot1-eGFP) were inserted into *N. benthamiana* leaves via agroinfiltration along with p19. The plants were incubated for 48 h in a growth room with a 16-h light (250 μmol/m²/s)/8-h dark cycle. The *N. benthamiana* leaves were ground in liquid nitrogen. The ground material was resuspended in extraction buffer (50 mM Tris-MES, pH 8.0, 0.5 M sucrose, 1 mM MgCl₂, 10 mM EDTA, 5 mM DTT, 1 mM PMSF, and Complete Mini protease inhibitor cocktail tablets). The extracts were centrifuged at $13,400 \times g$ for 15 min at 4 °C and the supernatants were collected for the Co-IP assays. We used GFP beads (SMART) and Flag beads (HuiOu Biotech) for the immunoprecipitation. For the immunoblot analysis, proteins were separated in 6%, 8%, or 10% SDS-PAGE gels and electroblotted to a pretreated polyvinylidene fluoride membrane at 120 V for 90 min. The membrane was blocked using TBS buffer containing 5% skimmed milk powder for 2 h at room temperature. The membrane was then sequentially incubated with the primary antibody at 4 °C overnight and then with the secondary antibody diluted in TBS buffer containing 1% skim milk powder for 1 h at room temperature. The anti-GFP (Abmart, CAT.M20004), anti-Myc (Abmart, CAT.M2000M), and anti-Flag (HuiOu Biotech, CAT.HOA012FL01) antibodies were used.

### Detergent-resistant membrane isolation
To isolate detergent-resistant membranes, the indicated constructs were expressed in agroinfiltrated *N. benthamiana* leaves along with p19. The transformed plants were incubated in a growth chamber for 2 days. The leaves were collected for the extraction of detergent-resistant membranes using the Minute™ Plant Lipid Raft Isolation Kit (Invent Biotechnologies). The isolated lipid rafts were resuspended in 100–200 μL Minute™ Denaturing Protein Solubilization Reagent (Invent Biotechnologies) for western blot analysis.

### Recombinant protein expression and purification
The pET28a, pHGWA, and pMGWA vectors were used to generate the pET28a-NFR1-CD, pHGWA-NFR5-CD, pMGWA-RinRK1-CD, and pMGWA-Flot1 plasmids, which were inserted into *E. coli* Rosetta cells for the expression of 6His- and MBP-tagged recombinant proteins. The transformed *E. coli* cells were grown overnight in an LB liquid medium at 37 °C. The overnight culture was used to inoculate fresh LB liquid medium (1:100 dilution). The bacteria were grown for 3–4 h at 37 °C and then 0.1 mM IPTG was added to induce protein expression. The culture was incubated at 20 °C for another 10 h. The 6His- or MBP-tagged recombinant proteins were purified using Ni-Sepharose beads (CAT.SA005005, SMART) and Amylose Resin (CAT.SA026010, SMART), respectively.

### In vitro kinase assay
Purified proteins were incubated at 30 °C for 30 min in 20 μL kinase reaction buffer containing 50 mM HEPES-NaOH (pH 7.5), 10 mM MgCl₂, 1 mM DTT, and 1.45 mM ATP. The kinase reaction was quenched by adding SDS-PAGE loading buffer and then the NFR1, NFR1[Y429F], and NFR1[T481A] kinase activities were analyzed using the anti-pTyr (GeneScript, CAT.A01819) or anti-pSer/Thr (ECM Biosciences, CAT.PP2551) antibodies. To evaluate whether NFR1 can phosphorylate Flot1, 1 μg purified recombinant His-NFR1-CD was incubated with 1 μg purified recombinant MBP-Flot1. The samples were subsequently analyzed using SDS-PAGE gels and was stained with Coomassie blue.

### Confocal microscopy
The Leica TCS SP8 confocal microscope and spinning disk confocal microscope (REVOLUTION XD, ANDOR) were used in this study. Images of cells expressing single fluorophores were acquired using a 20×/0.75 immersion objective for *N. benthamiana* leaves or a 60×/1.2 water immersion objective for root-hair membranes. The following excitation/emission settings were used: GFP (488 nm/498–540 nm), mCherry/DsRed (561 nm/570–610 nm), and mTag (405 nm/415–454 nm). Samples co-expressing two fluorophores were examined using the sequential mode between frames. Images were processed using the ImageJ software (http://rsbweb.nih.gov/ij/). Typical exposure times were 200–500 ms for GFP and 500 ms for DsRed and mCherry.

The sterol-disrupting reagent MβCD (Sigma Aldrich) was dissolved in deionized water to prepare a 200 mM stock solution. Transgenic plants on FP plates were incubated with 25 mM MβCD for 60 min and then observed under confocal microscopy. The MβCD treatments were carried out as previously described[60].

The ratio of the signal density at the root-hair tip to the signal density at the root-hair trunk was determined as previously described[22]. Spinning disk confocal microscopy was performed using a 60×/1.2 water immersion objective to image root-hair membranes. Z-projections of root hairs were created from approximately 2–45 images, taken at increments ranging from 0.5 μm to 2 μm. The stacks were processed using ImageJ software. The final representative image in the main figures shows a single optical section. For each construct, we selected at least 15 typical images, as described in the legends, to measure and calculate the fluorescence ratio. The fluorescence ratio was determined by measuring the signal density on the root-hair tip to the root-hair trunk cell surface. The calculation involved the following steps. (1) the average signal intensity of the root-hair tip was measured by drawing an ellipse along the root-hair tip membrane and using the ImageJ Measure function to determine the average intensity. (2) The signal intensity of the root-hair trunk was measured by drawing an ellipse along the trunk membrane (fluorescence value measured on each side of the trunk membrane). (3) The average intensity of the root-hair tip signal was divided by the root-hair trunk signal density (mean value of fluorescence of two side trunk membranes) to calculate the signal density ratio.

To calculate the proportion of fluorescence signal accumulation in root-hair tips, we cut off the susceptible area of transgenic roots of the same length and placed them on a slide. The fluorescence of root hairs was then observed under spinning disk confocal microscopy. The statistical value's denominator was the total number of observed fluorescent root hairs, while the numerator was the value of fluorescence accumulation at the root-hair tips, thereby characterizing the proportion of fluorescence accumulation at the root-hair tips.

For all samples, the roots derived from at least two independent transformations were examined, with images of 5–10 roots/root segments captured per transformation.

### Quantification and statistical analysis
For the phenotypic analysis, at least 10 individual plants were examined per accession. For the qRT-PCR analysis, tissue samples

from 8–10 individual plants were pooled and three technical replicates were examined. All data are presented herein as the mean ± standard deviation and were compared using One-way ANOVA, Two-way ANOVA or two-tailed, two-sample Student's $t$-tests (*$P < 0.05$, **$P < 0.01$) whenever appropriate. The statistical analysis for each experiment is described in the figure legends.

## Reporting summary

Further information on research design is available in the Nature Portfolio Reporting Summary linked to this article.

## Data availability

All data are available in the main text or the supplementary materials. Source data are provided with this paper.

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

## Acknowledgements

We thank Prof. Thomas Ott (University of Freiburg, Germany) for providing the CubG-LjSYMREM construct, Prof. Jeremy Murray (CEMPS, China) for helpful comments on the manuscript. Dr. Wenjuan Cai (CEMPS, China) for help with the microscopic analysis, and Dr. Ertao Wang and Dapeng Wang for help with the phosphorylation assay. This work was funded by the CAS Project for Young Scientists in Basic Research (YSBR-011), the Strategic Priority Research Program of the Chinese Academy of Sciences (XDB27040208), the Program of Shanghai Academic/Technology Research Leader (21XD1403900) and Shanghai Natural Science Fund (21ZR1471100).

## Author contributions

F.X. designed the experiments and supervised the study. N.Z., X.L., Z.Z. and J. L. performed the research. N.Z. and F.X. drafted the manuscript, and A. D. edited the manuscript. All authors read and approved the manuscript.

## Competing interests

The authors declare no competing interests.
