## [Peer Review File · Nature Communications]

RinRK1 Enhances NF Receptors Accumulation in Nanodomain-like structures at root hair tipReviewer #1 (Remarks to the Author):

This is a very interesting study that supplies several missing pieces of information in the story of Lj Nod factor receptor-interacting proteins. I think it will be of great interest to the symbiotic nitrogen fixation field.

Major points: 1) multiple lines of evidence for the interaction of RinRK1 with NFR1 and with NFR5 in *Lotus japonicus*. 2) BiFC data showing interaction between NFR1-RinRK1 and NFR5-RinRK1 in the plasma membrane of root hairs. 3) evidence that Flot1 (LotjaGi1g1v038900) is the major flotillin induced in root hairs and expressed in nodules. Also evidence that Flot1 interacts with RinRK1, NFR1 and NFR5. 4) evidence that both a flot1 mutant and Flot1 RNAi knockdown have a higher % of aberrant infection threads than wild type Lj. 5) Evidence that NFR1 Y428 (or 429? Numbering changes) is essential for Nfr1 function, and that NFR1 Y428 is autophosphorylated. My main criticism is that there is not enough information to evaluate the Ftip/Ftrunk GFP-localization assays in Figures 3, 4, 5, and 8. It may be possible for the authors to address this simply by providing more information about image collection and fluorescence quantification. The authors rely heavily on these GFP localization assays for their claims that the membrane proteins are redirected to the tip of the root hair upon inoculation with *M. loti* or with Nod factor. The authors cite reference 22 for the protocol for determining the fluorescence tip/trunk ratio. However, this reference provides very detailed steps for collection of z-sections, and exact steps for quantification of fluorescence in ImageJ. The authors should provide more detail on their procedure. This is critical, because they 'detect' an increase in Ftip/Ftrunk ratio in exactly all the conditions in which *M. loti* or purified NF induces a root hair response. These are also exactly the conditions in which the root hair tip swells and/or deforms and abolishes the straight focal planes of the root hair. There simply isn't enough information to evaluate these claims without knowing how the images were collected and quantified.

I think that the authors have actually under-emphasized their identification of the 'missing Flotillin' in Lj NFR interactions. This could be pushed more to the foreground of the manuscript, with some of the panels from Figure S5 and S6 moved to the main text.

Additional points:

Figure 1, The data is quite convincing for the interaction between the extracellular domains of RinRK1-NFR5 and RinRK1-NFR1. I'm not sure it is worthwhile to show lane 3 of the Western in Figure 1D. It appears that this provides little information about a possible interaction between RinRK1 and EPR3, since the IP of RinRK1 is poor.

Figure 2, The BiFC data is convincing for the interaction of RinRK1 with NFR1 and with NFR5, but I have the same issue re the Ftip/Ftrunk ratio that I described above for the GFP fluorescence.

Figure 3, 4, and 5 are all subject to my main criticism, described above.

Figure S5, S6—I really think some of the info on identification of Flot1 should be moved to main text.

Figure 6—The flot1-1 mutant has a reduced number of infection foci and ITs relative to wild type Lj. In contrast, in Figure S8, the uncomplemented flot1-1 mutant does not have a reduced number of infection foci or ITs. The authors should address this difference.

Figure 7b, In the top panel of the Western, it isn't clear what detection is shown since it says WB probe is 'MYC, or Flag' and both a Flag-tagged and a MYC-tagged protein have been included in lanes 2 and 4.

Figure 8, Same issue with GFP fluorescence described above.

Text lines 158-160 'The RinRK1 homologs AT5g1659037,38 and At2G2673039 in *A. thaliana* are present in the nanodomains of detergent-resistant proteasomes' Please rephrase this. I don't think 'proteasomes' is the word you intended here.

Lines 268-269 states '...in the flot1-1 mutant there were significantly fewer infection foci 269 and ITs; there were also more abnormal ITs in which infection by *M. loti*...'

Please make this consistent with the Figure 8 legend. Rather than 'more abnormal ITs' the figure shows more aberrant ITs as fraction or % of total ITs.

Reviewer #2 (Remarks to the Author):

The manuscript entitled "RinRK1 promotes accumulation of Nod Factor receptors in nanodomain enhancing rhizobial infection of legume roots" reports in the *Lotus japonicus* legume that the RinRK1 protein interacts both with the NFR1 and NFR5 receptors and with a flotillin, Flot1, in the root hair tip, promoting the relocalisation of the protein complex in nanodomain-like domains in response to symbiotic conditions induced by the rhizobium bacteria or by Nod factor signaling molecules.

This study confirms results previously obtained in *M. truncatula* where flotillins were shown to relocate in nanodomains with Nod factor receptors and to be linked to rhizobial infections in root hairs. Most of the results are based on correlations between the different protein accumulation domains in the root hair tip. In addition, the fact that mutants that have lost the ability to respond to symbiotic conditions (either rhizobium or Nod factors) fail to accumulate the receptor complex in nanodomains, was somewhat anticipated as these mutants cannot perceive the Nod factor inducing signal.

To gain more functional understanding, a FLOT1 mutant and RNAi approach were generated, which confirmed results obtained in other legumes about the role of flotillins in rhizobial infection in root hairs. This phenotype parallels the RinRK1 phenotypes, and further results suggest that the different partner of the receptor complex require each other's to allow the relocation of the complex in nanodomains in response to a symbiotic induction. The *nfr5* receptor mutant was however not used to test its relevance for Flot1, RinRK and NFR1 relocalisation.

In a final part, the relevance of the kinase activity of NFR1 for its symbiotic nanodomain accumulation in root hairs was tested, but surprisingly not the role of the RinRK kinase activity in the complex formation. A Flot1-Y161F variant was used without defining clearly the goal of generating this mutation. On a related line, the use of the NFR1 phosphorylation site variant NFR1-Y429F is not clearly justified (dominant phosphomimic?). In addition, the conclusion based on a negative result that FLOT1 is not a substrate of NFR1 should be downsized. The final conclusion sentence of the result section (l. 377) is unclear to me, and in the discussion line 419, the "biological functions" mentioned are too general.

In the IP figures, some inputs are missing (Flag in Fig1, MYC and GFP in Fig 7). What means "WB: MYC or Flag" in the Fig 7b? In the Fig 2 f, 2i, 3b, 4h, 4p, and 5k, the NF treatment in the WT calls for testing this condition in the mutants. Alternatively, show graphs as in Fig 8, without the Nod factor treatment and the *nodC* mutant? In the fig 6g and j, as well as from the Fig S8e or S12, the infection phenotypes are not always consistent, so this should be explained. Ri should be defined in the legend of the Fig 6. Source data for western blots should be included in a dedicated Supplementary Figure.

Reviewer #3 (Remarks to the Author):

In their manuscript, Zhou et al characterized the membrane-resident receptor RinRK1 as a component interacting with the Nod Factor receptors NFR1 and NFR5. Using the flotillin 1 mutant, they provide data indicating that this interaction depends on FLOT1. Furthermore, they claim that accumulation of the NFRs at the root hair tip is RinRK1- and FLOT1-dependent and relate this to a direct interaction with FLOT1. In a last set of experiments they assessed a putative Tyr phosphorylation activity of the NFR1 kinase domain.

In general, the paper provides a large amount of data. However, I have some general concerns about controls, conclusions and the experimental design that I would like to see addressed. I listed them in the order of appearance.

1. Figure 1b,d: Please indicate more clearly in the figure that only the EDs were used rather than the full-length protein (as done in Suppl. Fig. 1 for the CD).
2. Suppl. Fig. 1: An expression control (Western Blot) needs to be provided for all combinations of this experiment as the absence of interactions might also be caused by the lack of protein itself.
3. The BiFC assays in Fig. 2 and Suppl. Fig. 2 lack essential negative controls as this method is highly prone to result in false positive results especially when proteins are constitutively over-expressed as done here. The authors should use EPR3 as they did in the yeast assays and need to confirm presence of the proteins especially when no fluorescence is observed.
4. General comment to Fig. 2 and similar ones below: I would like to ask the authors to clarify whether the numbers on the panels are from one biological replicate or of all three replicates together? For the Tip/Trunk quantifications: What were the selection criteria for the root hairs that were scored? Given the large number of non-responding root hairs, I am a bit puzzled about the comparably small error bars, especially when considering for example, that only 58/155 root hairs showed that phenomenon (Fig. 2b). For my understanding, it would be extremely important to select the scored root hairs on features such as morphological changes, nuclear movement, etc and then test how many of those responsive root hairs show the tip accumulations. I would also like to comment on the statistics. In general, the classical bar diagram is outdated as it is not a comprehensive representation of the data. Same accounts for the t-test, which was used in most cases. There have been ample reports showing that such analysis is not robust enough (compare to e.g. Brady et al., 2015, Plant Cell)
5. line 159: Use the same style when indicating gene names.
6. line 160: The authors need to explain, what they mean with "detergent-resistant proteasomes". In addition, there has been a whole lot of literature indicating that detergent-resistant membrane fractions do not represent membrane nanodomains. It has been actually a difficult task to identify transmembrane proteins (especially when being over-expressed) that are not enriched in this fraction. I would therefore strongly recommend to carefully revise all of these statements and even to consider removing these data. These experiments certainly do not justify the use of the term "nanodomain" in the title of the manuscript.
7. Fig. 3 b-f: Personally, I fear that the authors over-interpret their data here. Application of isolated NFs or NF-secreting rhizobia will result in severe changes in root hair structures as one can also see in panel 3d. Consequently, protein secretion is mainly focused towards the root hair tip under these conditions and would result in an accumulation of most proteins there. I believe that the same criticism applies to the data shown in Fig. 4. Furthermore, the authors need to state more explicitly, which root hairs they scored (see above). Did all root hairs (289 in d and 164 in f) showed unambiguous signs of root hair deformations?
8. line 189: Why did the authors use different over-expression promoters for different experiments?
9. Supplementary Fig. 5a: From the phylogenetic analysis presented here, it is very difficult to identify orthologs/paralogs in between the species as all proteins still cluster rather by species than by the protein subfamilies. Furthermore, the resolution of the nodes is poor and does not allow firm statements.
10. line 227: This "punctate co-localization" cannot be validated based on these images. First, the resolution is far too low. Second, it is extremely difficult if not impossible to judge on such punctates in lateral membranes given the poor resolution of confocal microscopes along this axis. The authors should provide images from surfaces or deconvoluted single plane z-stack images. In addition, these observations (as all other ones) need to be quantified to match current cell biological standards.
11. Fig. 5. Again, accumulation of FLOT1 may be a simple result of increased membrane/protein delivery to this site. Why do the authors now show root hair tips without deformation in panel 5b? Why is the number of tip-accumulated FLOT1 so much higher than the values observed for the receptors?
12. Fig. 6. The phenotyping data between the *flot1-1* and the RNAi constructs are very different when scoring infection events/plants but the authors don't comment on this.
13. Fig. 7a: There are no proper specificity controls for the Y2H assay using another kinase

domain. This should be emphasized as other studies failed to demonstrate a reliable interaction between symbiotic receptors and flotillins (e.g. Haney and Long, 2009; Haney et al., 2011).

14. Fig. 7b: Are the authors sure about the molecular weight indicated for FLOT1? Isn't much smaller compared to the NFRs?

15. line 312/Suppl. Fig. 9b: There are also punctate patterns with NFR5 in this panel. None of these observations is quantified. Why are these patterns not observed in Lotus root hairs?

16. Fig. 8: Why do the authors observe such as strong band shift for the mutated kinase domains? Could they please indicate the MW in the anti-His WB?

REVIEWER COMMENTS

Reviewer #1 (Remarks to the Author):

This is a very interesting study that supplies several missing pieces of information in the story of Lj Nod factor receptor-interacting proteins. I think it will be of great interest to the symbiotic nitrogen fixation field.

Major points: 1) multiple lines of evidence for the interaction of RinRK1 with NFR1 and with NFR5 in *Lotus japonicus*. 2) BiFC data showing interaction between NFR1-RinRK1 and NFR5-RinRK1 in the plasma membrane of root hairs. 3) evidence that Flot1 (LotjaG1g1v038900) is the major flotillin induced in root hairs and expressed in nodules. Also evidence that Flot1 interacts with RinRK1, NFR1 and NFR5. 4) evidence that both a *flot1* mutant and Flot1 RNAi knockdown have a higher % of aberrant infection threads than wild type Lj. 5) Evidence that NFR1 Y428 (or 429? Numbering changes) is essential for Nfr1 function, and that NFR1 Y428 is autophosphorylated.

Response: We thank Reviewer #1 for her/his positive feedback on our work. We appreciate your time and effort invested in providing us with a critical review, which has been immensely valuable in enhancing the quality of our manuscript. We believe the revised manuscript will be significantly improved with your great support.

My main criticism is that there is not enough information to evaluate the Ftip/Ftrunk GFP-localization assays in Figures 3, 4, 5, and 8. It may be possible for the authors to address this simply by providing more information about image collection and fluorescence quantification. The authors rely heavily on these GFP localization assays for their claims that the membrane proteins are redirected to the tip of the root hair upon inoculation with *M. loti* or with Nod factor. The authors cite reference 22 for the protocol for determining the fluorescence tip/trunk ratio. However, this reference provides very detailed steps for collection of z-sections, and exact steps for quantification of fluorescence in ImageJ. The authors should provide more detail on their procedure. This is critical, because they 'detect' an increase in Ftip/Ftrunk ratio in exactly all the conditions in which *M. loti* or purified NF induces a root hair response. These are also exactly the conditions in which the root hair tip swells and/or deforms and abolishes the straight focal planes of the root hair. There simply

isn't enough information to evaluate these claims without knowing how the images were collected and quantified.

Response: Thanks to the reviewer for bringing this issue to our attention. We acknowledge that we did not provide a clear explanation of the procedure in our initial manuscript. We have now included a detailed description of the procedure in the Methods section (Line 692-715). We believe that this clarification will greatly enhance the understanding of the methodology and improve the overall quality of our manuscript.

I think that the authors have actually under-emphasized their identification of the 'missing Flotillin' in Lj NFR interactions. This could be pushed more to the foreground of the manuscript, with some of the panels from Figure S5 and S6 moved to the main text.

Response: Thank you for your suggestion. We have moved the Flotillin phylogenetic tree and expression pattern to the main Fig. 5a, 5b.

Additional points:

Figure 1, The data is quite convincing for the interaction between the extracellular domains of RinRK1-NFR5 and RinRK1-NFR1. I'm not sure it is worthwhile to show lane 3 of the Western in Figure 1D. It appears that this provides little information about a possible interaction between RinRK1 and EPR3, since the IP of RinRK1 is poor.

Response: Thank you for your suggestion. We have removed the lane 3 from Fig 1d, which is now labeled as Fig. 1c. In addition, we conducted the Co-IP assay in parallel to address any potential concerns related to their interaction (Fig 1c).

Figure 2, The BiFC data is convincing for the interaction of RinRK1 with NFR1 and with NFR5, but I have the same issue re the Ftip/Ftrunk ratio that I described above for the GFP fluorescence.

Figure 3, 4, and 5 are all subject to my main criticism, described above.

Response: Thank you for your comments. We have now provided a detailed description of the procedure in the Methods section (Line 692-715).

Figure S5, S6—I really think some of the info on identification of Flot1 should be moved to main

text.

Response: Thank you for your suggestion. We have moved the Flotillin phylogenetic tree and expression pattern to the main figure (Fig. 5a, 5b).

Figure 6—The *flot1-1* mutant has a reduced number of infection foci and ITs relative to wild type *Lj*. In contrast, in Figure S8, the uncomplemented *flot1-1* mutant does not have a reduced number of infection foci or ITs. The authors should address this difference.

Response: We're apologize for the confusion caused, as similar concerns were raised by other two reviewers as well. In response to this, we conducted further experiments to address the issue. We confirmed a complementation assay using either *LjUb* or *35S* promoter driven *Flot1*(*p35S:Flot1* or *pUb:Flot1*, which was used for Flot1 protein subcellular location or BiFC) in the *flot1-1* mutant. We then assessed the infection events per centimeter at 5 dpi. The results demonstrated that *flot1-1* (*EV/flot1-1*) exhibited few infection events compared to the wild type (*EV/WT*), whereas *Flot1/flot1-1* transgenic roots displayed more infection events than *EV/flot1-1* and were similar with *EV/WT*. These findings have been included in Fig. S5.

Furthermore, we validated the *flot1* infection events by using CRISPR/Cas9 knock-out *Flot1*, as depicted in in Fig. 5j and Fig. S5c. The results revealed similar infection defects as observed in the *flot1-1* mutant. Collectively, these findings confirm the essential role of *Flot1* in rhizobial infection.

Figure 7b, In the top panel of the Western, it isn't clear what detection is shown since it says WB probe is 'MYC, or Flag' and both a Flag-tagged and a MYC-tagged protein have been included in lanes 2 and 4.

Response: We're apologize for the confusion caused. We have labelled clearly the Fig. 7b now. Thank you for bringing this to our attention.

Figure 8, Same issue with GFP fluorescence described above.

Response: Point taken, the detail description was update in our methods now.

Text lines 158-160 'The RinRK1 homologs AT5g1659037,38 and At2G2673039 in *A. thaliana* are present in the nanodomains of detergent-resistant proteasomes' Please rephrase this. I don't think

‘proteasomes’ is the word you intended here.

Response: We apologize for the mistake. The correct term should be “detergent-resistant membranes (DRMs)”. We have made the necessary correction and rephrased it in Line 151-152. Thank you for pointing out the error.

Lines 268-269 states ‘...in the *flot1-1* mutant there were significantly fewer infection foci 269 and ITs; there were also more abnormal ITs in which infection by *M. loti*...’

Please make this consistent with the Figure 8 legend. Rather than ‘more abnormal ITs’ the figure shows more aberrant ITs as fraction or % of total ITs.

Response: We apologize for the confusion caused by our previous description. In our study, we observed an increased number of abnormal infection events in *Flot1* RNAi lines, which is a similar phenotype reported in *MtFlot4* RNAi (Haney and Long, 2010, PNAS). However, in the case of *Ljflot1-1*, we found that there were fewer infection foci and infection threads (ITs), but the number of abnormal ITs was not significantly higher compared to the wild type. The only notable difference was a higher ratio of abnormal to total infection events, this could be a consequence of the lower total infection events in the mutant. We further validated this phenotype by conducting *Flot1* CRISPR/Cas9 knock-out experiments, as illustrated in the Fig. 5j. Moreover, we have concerns that the *Flot1* RNAi line may have affected other Flotilins, we decided to remove the *Flot1* RNAi results and revised the corresponding description in Line 249-253 to reflect this decision.

Reviewer #2 (Remarks to the Author):

The manuscript entitled “RinRK1 promotes accumulation of Nod Factor receptors in nanodomain enhancing rhizobial infection of legume roots” reports in the *Lotus japonicus* legume that the RinRK1 protein interacts both with the NFR1 and NFR5 receptors and with a flotillin, Flot1, in the root hair tip, promoting the relocalisation of the protein complex in nanodomain-like domains in response to symbiotic conditions induced by the rhizobium bacteria or by Nod factor signaling molecules.

This study confirms results previously obtained in *M. truncatula* where flotillins were shown to

relocate in nanodomains with Nod factor receptors and to be linked to rhizobial infections in root hairs. Most of the results are based on correlations between the different protein accumulation domains in the root hair tip. In addition, the fact that mutants that have lost the ability to respond to symbiotic conditions (either rhizobium or Nod factors) fail to accumulate the receptor complex in nanodomains, was somewhat anticipated as these mutants cannot perceive the Nod factor inducing signal.

To gain more functional understanding, a FLOT1 mutant and RNAi approach were generated, which confirmed results obtained in other legumes about the role of flotillins in rhizobial infection in root hairs. This phenotype parallels the RinRK1 phenotypes, and further results suggest that the different partner of the receptor complex require each other's to allow the relocation of the complex in nanodomains in response to a symbiotic induction. The *nfr5* receptor mutant was however not used to test its relevance for Flot1, RinRK and NFR1 relocalisation.

Response: We appreciate the reviewer's summary and supports for our study. As suggested, we have conducted new experiments and the relocalisation of Flot1, RinRK1 and NFR1 in *nfr5-3* (Fig. 2i, j; Fig. 3f, g; Fig. 6k, l), and also observed the relocalisation of NFR5 in *nfr1-1* (Fig. 3p, q). Taken together, these results demonstrate that the relocalisation of NFR1, NFR5, Flot1, and RinRK1 at the root hair tips induced by rhizobia requires NF receptors, RinRK1 and Flot1.

In a final part, the relevance of the kinase activity of NFR1 for its symbiotic nanodomain accumulation in root hairs was tested, but surprisingly not the role of the RinRK kinase activity in the complex formation. A Flot1-Y161F variant was used without defining clearly the goal of generating this mutation. On a related line, the use of the NFR1 phosphorylation site variant NFR1-Y429F is not clearly justified (dominant phosphomimic?). In addition, the conclusion based on a negative result that FLOT1 is not a substrate of NFR1 should be downsized. The final conclusion sentence of the result section (l. 377) is unclear to me, and in the discussion line 419, the "biological functions" mentioned are too general.

Response: Sorry we may not have explained it clearly before. RinRK1 is an atypical kinase that lacks auto-phosphorylation activity (Li et al., 2019, Plant Physiology). This information is included in the introduction (Line 99-101).

In mammal studies found that Fyn kinase can directly phosphorylate the critical Tyr residues Y160 in Flot1 and Y163 in Flot2, and mutation of these Tyr residues affects the subcellular localization and function of Flot proteins. Therefore, we aim to investigate whether Flot1 can be phosphorylated by NFR1, and whether the Flot1 Y161 residue (the corresponding site of mammal Y160 in Flot1 and Y163 in Flot2) is important for its relocation to root hair tips induced by NF and complement *flot1-1* phenotype.

We observed that the NFR1^{Y429F}, which disrupts a crucial Tyr phosphorylation site in NFR1, diminished NFR1's Tyr phosphorylation activity but has no effect on the *M. loti* induced NFR1 accumulation at root hair tips. Although we did not detect phosphorylation of Flot1 by NFR1 in our *in vitro* experiment, we cannot draw a conclusion based solely on negative results. To avoid overinterpretation, we have weakened this conclusion and removed the results regarding Flot1^{Y161F} mutation, as it cannot prove this residue is the actual Tyr phosphorylation site of Flot1.

We apologize for the confusion caused by our previous description, the “biological functions” there means “symbiotic interaction with rhizobia”. We have made the necessary correction and rephrased it in Line 389-393 and Line 447-449.

In the IP figures, some inputs are missing (Flag in Fig 1, MYC and GFP in Fig 7). What means “WB: MYC or Flag” in the Fig 7b? In the Fig 2 f, 2i, 3b, 4h, 4p, and 5k, the NF treatment in the WT calls for testing this condition in the mutants. Alternatively, show graphs as in Fig 8, without the Nod factor treatment and the *nodC* mutant? In the fig 6g and j, as well as from the Fig S8e or S12, the infection phenotypes are not always consistent, so this should be explained. Ri should be defined in the legend of the Fig 6. Source data for western blots should be included in a dedicated Supplementary Figure.

Response: We apologize for any confusion caused earlier, and we appreciate your clarification.

- 1) We have now added the Co-IP inputs for Fig. 1 and Fig. 7, and relabeled the Fig 7b.
- 2) In Fig 2, 3, 4 and 5 (now updated to Fig 2, 3, 4 and 6), our objective was to investigate that the rhizobia induced NFR1, NFR5, RinRK1, Flot1, or NFR1/5-RinRK1 interaction at root hair tips was dependent on NFs. To address this, we utilized purified NF or a *nodC* mutant, which cannot produce NF, to validate our findings. Based on our research, we found that NFs are sufficient to induce the relocation of NFRs to root hair tips. Based on our findings, in Fig. 8, we want further

to analyze whether the phosphorylation sites of NFR1 were required for rhizobia-induced relocalisation. We believe that the treatment of *M. loti* can be trusted and there is no need to use NF or *nodC* mutant again.

3) We're apologized for the confusion caused for the *Flot1* mutant phenotype, as similar concerns were raised by other two reviewers as well. In response to this, we conducted further experiments to address the issue. We confirmed a complementation assay using either *LjUb* or *35S* promoter driven *Flot1* (*p35S:Flot1* or *pUb:Flot1*, which was used for *Flot1* protein subcellular location or BiFC) in the *flot1-1* mutant. We then assessed the infection events per centimeter at 5 dpi. The results demonstrated that *flot1-1 (EV/flot1-1)* exhibited few infection events compared to the wild type (*EV/WT*), whereas *Flot1/flot1-1* transgenic roots displayed more infection events than *EV/flot1-1* and were similar with *EV/WT*. These findings have been included in Fig. S5. Furthermore, we validated the *flot1* infection events by using CRISPR/Cas9 knock-out *Flot1*, as depicted in Fig. 5j and Fig. S5c. The results revealed similar infection defects as observed in the *flot1-1* mutant. Collectively, these findings confirm the essential role of *Flot1* in rhizobial infection.

Regarding to *Flot1* RNAi, we observed an increased number of abnormal infection events in *Flot1* RNAi lines, which is a similar phenotype reported in *MtFlot4* RNAi (Haney and Long, 2010, PNAS). However, in the case of *Ljflot1-1*, we found that there were fewer infection foci and infection threads (ITs), but the number of abnormal ITs was not significantly higher compared to the wild type. The only notable difference was a higher ratio of abnormal to total infection events, this could be a consequence of the lower total infection events in the mutant. We further validated this phenotype by conducting *Flot1* CRISPR/Cas9 knock-out experiments, as illustrated in the Fig. 5j. Moreover, we have concerns that the *Flot1* RNAi line may have affected other Flotillins, we decided to remove the *Flot1* RNAi results and revised the corresponding description in Line 249-253 to reflect this decision.

Source data for all figures were included now.

Reviewer #3 (Remarks to the Author):

In their manuscript, Zhou et al characterized the membrane-resident receptor RinRK1 as a component interacting with the Nod Factor receptors NFR1 and NFR5. Using the flotillin 1 mutant,

they provide data indicating that this interaction depends on FLOT1. Furthermore, they claim that accumulation of the NFRs at the root hair tip is RinRK1- and FLOT1-dependent and relate this to a direct interaction with FLOT1. In a last set of experiments they assessed a putative Tyr phosphorylation activity of the NFR1 kinase domain.

Response: We appreciate the reviewer's summary and supports for our study.

In general, the paper provides a large amount of data. However, I have some general concerns about controls, conclusions and the experimental design that I would like to see addressed. I listed them in the order of appearance.

1. Figure 1b, d: Please indicate more clearly in the figure that only the EDs were used rather than the full-length protein (as done in Suppl. Fig. 1 for the CD).

Response: Point taken. The extracellular domain (ED) were labelled in Fig. 1b, as well as Fig. S1.

2. Suppl. Fig. 1: An expression control (Western Blot) needs to be provided for all combinations of this experiment as the absence of interactions might also be caused by the lack of protein itself.

Response: Thank you for your suggestion. The western blot results for Fig S1 were included now.

3. The BiFC assays in Fig. 2 and Suppl. Fig. 2 lack essential negative controls as this method is highly prone to result in false positive results especially when proteins are constitutively over-expressed as done here. The authors should use EPR3 as they did in the yeast assays and need to confirm presence of the proteins especially when no fluorescence is observed.

Response: We agree BiFC assays can potentially produce false positive results, but we conducted several experiments, including Y2H, Co-IP, and now include split Luciferase complementation assays to demonstrate the interaction between RinRK1 and NFR1 or NFR5. We further trying to analyze their interaction in *Lotus* roots before and after rhizobia inoculation. However, expressing two individual constructs simultaneously in the same roots using hairy roots proved challenging. To overcome this, we employed Golden gate cloning to generate BiFC constructs for NFR1-RinRK1 and NFR5-RinRK1, as well as Flot1 and symbiotic receptors. This approach can assembly multiple

DNA fragments in a recipient vector, which has been successfully used in previous studies, such as the work by Liu et al., (2019, *Nat Commu.* 10:2848 | <https://doi.org/10.1038/s41467-019-10029-y>). To address the negative control for BiFC, we made efforts to validate the interaction by attempting to create a BiFC construct for RinRK1 and EPR3. Unfortunately, we encountered difficulties during several attempts, prompting us to utilize split luciferase complementation method - a widely used approach for examining protein-protein interactions. The results obtained from this split luciferase assay confirmed the findings from our Y2H and Co-IP experiments. Regrettably, we were unable to investigate the protein levels through western blot analysis due to the unavailability of native antibodies for RinRK1 and EPR3. Additionally, the Luc antibody yield unsatisfactory results.

We apologize for the mistake we made previously. Our objective is to investigate the interaction between RinRK1 and NF receptors, MtLYK10, being a transmembrane LysM RLK, fulfills the criteria for an appropriate control. Since one of the students in our lab is working on MtLYK10 and we have multiple constructs related to MtLYK10, it made sense to use it as a negative control. Initially, when our student work on this gene, it was an orthologue named EPR3 in *L. japonicus*, so we used MtEPR3 that time. However, a study conducted on *M. truncatula* named this gene as MtLYK10 (Mailet et al., 2020, *Plant J.* doi: 10.1111/tpj.14625). Therefore, in this manuscript, we are using the name MtLYK10 instead of EPR3. We apologize for any confusion caused by the change in gene name, but we are following the naming convention used in the study conducted on *M. truncatula*.

4. General comment to Fig. 2 and similar ones below: I would like to ask the authors to clarify whether the numbers on the panels are from one biological replicate or of all three replicates together?

Response: We apologize for any confusion caused previously. All the numbers presented in the figure panels were obtained by combined data from two or three replicates. However, the fluorescence ratio between root hair tips and the trunk was analyzed using 10-20 individual root hairs. We have now made this clarification in the Methods section (Line 692-715) to provide clear and accurate information.

For the Tip/Trunk quantifications: What were the selection criteria for the root hairs that were scored? Given the large number of non-responding root hairs, I am a bit puzzled about the comparably small error bars, especially when considering for example, that only 58/155 root hairs

showed that phenomenon (Fig. 2b). For my understanding, it would be extremely important to select the scored root hairs on features such as morphological changes, nuclear movement, etc and then test how many of those responsive root hairs show the tip accumulations.

Response: Sorry for this confusion. We agree this quantification are important and this was raised by another reviewer, we have clarified this in our methods now (Line 692-715). However, as we clarified in response to subsequent comments, NF triggered root hair deformation and rhizobia (NF) induced accumulation of symbiotic receptors at root hair tips, were two distinct phenomena. Therefore, not all root hairs that accumulated symbiotic receptors or Flot1 displayed evident root hair deformation. This observation is consistent with previous studies in *M. truncatula* and *G. max*.

I would also like to comment on the statistics. In general, the classical bar diagram is outdated as it is not a comprehensive representation of the data. Same accounts for the t-test, which was used in most cases. There have been ample reports showing that such analysis is not robust enough (compare to e.g. Brady et al., 2015, Plant Cell)

Response: Thank you for your suggestion, we have made the necessary changes accordingly.

We have replaced the classical bar diagram with a separated scatter graph that includes bars to display each data point in Fig. 2, 3, 4, 6, 7, 8.

For certain experiments, such as Fig 5, 7, 8 and Fig. S2, S3, S4, our aim was to assess whether there were differences between the experimental group and the control group. To compare the mean difference between two samples, the t-test is applicable, and we believe that conducting a two-tailed student t-test analysis is sufficient for these cases. However, when comparing the means of three or more groups, the t-test is not an effective statistical test and may introduce bias into the results. Therefore, we have replaced the t-test with one-way analysis of variance (ANOVA) in Fig. 4 and two-way ANOVA in Fig. S5, S8. ANOVA can eliminate the probability of the first type error accumulation, avoiding rejecting a true null hypothesis, and increase the reliability of the analysis.

5. line 159: Use the same style when indicating gene names.

Response: We apologize for the mistake. We have corrected these two gene names with same style (Line 151-152).

6. line 160: The authors need to explain, what they mean with "detergent-resistant proteasomes".

In addition, there has been a whole lot of literature indicating that detergent-resistant membrane fractions do not represent membrane nanodomains. It has been actually a difficult task to identify transmembrane proteins (especially when being over-expressed) that are not enriched in this fraction. I would therefore strongly recommend to carefully revise all of these statements and even to consider removing these data. These experiments certainly do not justify the use of the term "nanodomain" in the title of the manuscript.

Response: We apologize for the mistake. The correct term should be “detergent-resistant membranes (DRMs)”. We have made the necessary correction and rephrased it in Line 152. Thank you for pointing out the error.

Regarding to the nanodomain, we would like to highlight the following points:

1) Except RinRK1 homologues were found to be enriched in DRMs. Additionally, we used a kit for isolation detergent-insoluble proteins to assess the presence of RinRK1 and Flot1 in *N. benthamiana* leaf. Our results demonstrated that both RinRK1 and Flot1 were enriched in the DRM fraction.

2) Flotillin serves as a commonly used markers for nanodomains in mammal as well as several plant species. Moreover, similar to MtFlot4, GmFlot2/4 and GmFWL1, we observed the accumulation of RinRK1 and Flot1 in root hair tips after rhizobia inoculation. We also detect puncta dots of RinRK1 in root hair tips either alone or NFR5-RinRK1 in wild type *L. japonicus* (see below), this was consistent with the observation in *Medicago* and soybean. Although the observation ratio was relatively low, this could be attributed to the low expression level.

3) To further support our findings, we combined with M β CD, a specific inhibitor for sterol-dependent membrane proteins, and observed a reduction in rhizobia-induced accumulation of

RinRK1-eGFP and Flot1-eGFP in root hair tips (Fig. 2m & Fig. 6m). Considering all these observations, we believe it is appropriate to include the term “nanodomain” in our manuscript, including in the title.

7. Fig. 3 b-f: Personally, I fear that the authors over-interpret their data here. Application of isolated NFs or NF-secreting rhizobia will result in severe changes in root hair structures as one can also see in panel 3d. Consequently, protein secretion is mainly focused towards the root hair tip under these conditions and would result in an accumulation of most proteins there. I believe that the same criticism applies to the data shown in Fig. 4. Furthermore, the authors need to state more explicitly, which root hairs they scored (see above). Did all root hairs (289 in d and 164 in f) showed unambiguous signs of root hair deformations?

Response: Yes, treatment with NF or rhizobia resulted in significantly changes, specifically referred to as root hair deformation (RHD). However, the main focus of our study was to observe the accumulation of symbiotic receptors induced by rhizobia and NF at the nanodomains located at root hair tips. It is important to note that NF induced RHD and the accumulation of proteins induced by rhizobia (or NF) at the nanodomains of the root hair tips are not the same thing. Studies conducted in *M. truncatula*, such as Haney et al (2011, Plant Cell) and Liang et al (2018, PNAS), have shown that LYK3, Flot4 and SymRem1 are not essential for NF-induced RHD but are involved in the formation of protein clusters in the PM nanodomains induced by rhizobia. In our study, both *rinrk1* and *flot1* mutants exhibited normal responses to NF-induced RHD. However, the accumulation of NFRs, Flot1 and RinRK1 at root hair tips in response to NF and these accumulations were dependent on RinRK1 and Flot1. Additionally, the ND inhibitor M β CD, reduced the accumulation of RinRK1 and Flot1 at root hair tips, although it did not affect NF induced RHD. Therefore, we maintain that we have not overinterpreted our data.

We apologize for not providing a clear explanation of the inflorescence ratio previously. We have now included a detailed description of the procedure in the Methods section (Line 692-715) to address this issue. Although we selected the susceptible zone for observation, not all of the observed root hairs exhibited clear root hair deformation.

8. line 189: Why did the authors use different over-expression promoters for different experiments?

Response: We originally use *p35S:RinRK1* construct but found that it could not rescue the infection phenotype of *rinrk1*. Therefore, we switched to the *pUb:RinRK1* construct, which was able to rescue *rinrk1* phenotype (Fig. S2). All other constructs used in this study were confirmed using complementation mutants to ensure their functionality.

9. Supplementary Fig. 5a: From the phylogenetic analysis presented here, it is very difficult to identify orthologs/paralogs in between the species as all proteins still cluster rather by species than by the protein subfamilies. Furthermore, the resolution of the nodes is poor and does not allow firm statements.

Response: We apologize for the previous inadequate phylogenetic analysis. Our phylogenetic tree was constructed according to the neighbor-joining method using MEGA7. We have conducted an additional phylogenetic analysis according to the Maximum Likelihood method using MEGA7, which was now included in Fig. 5a.

10. line 227: This "punctate co-localization" cannot be validated based on these images. First, the resolution is far too low. Second, it is extremely difficult if not impossible to judge on such punctates in lateral membranes given the poor resolution of confocal microscopes along this axis. The authors should provide images from surfaces or deconvoluted single plane z-stack images. In addition, these observations (as all other ones) need to be quantified to match current cell biological standards.

Response: Thank you for bringing this problem to our attention. We have rephased the sentence, "punctate co-localization" in plasma membrane were indicated by orange triangles and the co-localization fluorescence intensity was measured using Image J (Fig. S6).

11. Fig. 5. Again, accumulation of FLOT1 may be a simple result of increased membrane/protein delivery to this site. Why do the authors now show root hair tips without deformation in panel 5b? Why is the number of tip-accumulated FLOT1 so much higher than the values observed for the receptors?

Response: The Flotillin proteins are often utilized as markers for cholesterol-rich, detergent-resistant, membrane microdomains (or nanodomains). Similar observations have been reported in the case of MtFlot4 (Haney and Long, 2010, PNAS) and GmFlot2/4 (Qiao et al., 2017, Plant, Cell

& Environment), where polar location at root hair tips was observed after rhizobia inoculation. The corresponding figures illustrating these observations are provided below.

As we mentioned earlier, the RHD and nanodomain localization are distinct phenomena. Therefore, not all root hair exhibit RHD even if Flot1-eGFP accumulates at root hair tips. We choose an image that clearly depicted accumulation at root hair tips.

In our observation, it appears that Flot1 has a higher tip-accumulation at the root hair tips compared to other receptors. Flot1 belongs to the SPFH domain family, which includes the SPFH (also known as PHB) domain in the N-terminal. The SPFH domain has the ability to bind to lipids and assemble into membrane-bound oligomers, forming potential scaffolds. Studies in animals have shown that Flotillin-1 and Flotillin-2 are located in the microdomain of the plasma membrane, and their localization is mediated by the N-terminal SPFH domain. Additionally, previous research in *M. truncatula* (Liang et al, 2018, PNAS) has demonstrated that “SYMREM1 is recruited into nanodomains in a FLOT4-dependent manner, with FLOT4 serving as a central hub during primary nanodomain assembly”. Based on this knowledge, we speculate that the lipid binding ability of Flot1’s N-terminal region in addition to other unknown functions during primary nanodomain assembly may contribute to Flot1’s rapid response to rhizobia signals. This could explain why the number of tip-accumulated FLOT1 is higher compared to other receptors.

Fig. 3. FLOTs localize to membrane-associated puncta and become polar after inoculation. We generated A17 hairy roots expressing 35S:FLOT2:GFP or FLOT4:GFP driven by its native promoter. Transgenic roots were visualized by using a spinning-disk confocal microscope. (Scale bars: 15 μ m.) FLOT4:GFP and FLOT2:GFP are visibly punctate and evenly distributed in the membranes of uninoculated root hair cells. FLOT4:GFP puncta localize to root hair tips by 1 dpi, whereas the localization of FLOT2:GFP does not change upon inoculation. FLOT2:GFP is weakly polar in uninoculated epidermal cells (arrows) and remains polar on inoculation. At least 15 transgenic lines were observed for each treatment. Representative images are shown. Root hair images are overlays of 100 sections, taken at 0.2- μ m increments.

Figure 5. Subcellular localization of mCherry-GmFLOT2/4 fusion protein in soybean root hair cell in response to *Bradyrhizobium japonicum* inoculation. Soybean root hair cells were mock-inoculated (a, b, e and f) or inoculated with *B. japonicum* (c, d, g, h and i). One day (a-d) and 7 d (e-i) after inoculation, the transgenic root hair cells were observed under an epifluorescent confocal microscope. Compared to mock-inoculated conditions and similarly to GmFWL1 (Fig. 5), GmFLOT2/4 accumulation at the tip of the root hair cells in response to *B. japonicum* inoculation was observed as soon as 24 h after inoculation and continued 7 d after inoculation. White arrows highlight the accumulation of the mCherry-GmFLOT2/4 protein at the tip of the root hair cells upon *B. japonicum* inoculation. Supporting Information Fig. S6 shows the quantification of the mCherry signal in inoculated and mock-inoculated transgenic root hair cells.

12. Fig. 6. The phenotyping data between the *flot1-1* and the RNAi constructs are very different when scoring infection events/plants but the authors don't comment on this.

Response: We're apologize for the confusion caused for the *Flot1* mutant phenotype, as similar concerns were raised by other two reviewers as well. In response to this, we conducted further

experiments to address the issue. We confirmed a complementation assay using either *LjUb* or *35S* promoter driven *Flot1* (*p35S:Flot1* or *pUb:Flot1*, which was used for Flot1 protein subcellular location or BiFC) in the *flot1-1* mutant. We then assessed the infection events per centimeter at 5 dpi. The results demonstrated that *flot1-1* (*EV/flot1-1*) exhibited few infection events compared to the wild type (*EV/WT*), whereas *Flot1/flot1-1* transgenic roots displayed more infection events than *EV/flot1-1* and were similar with *EV/WT*. These findings have been included in Fig. S5. Furthermore, we validated the *flot1* infection events by using CRISPR/Cas9 knock-out *Flot1*, as depicted in Fig. 5j and Fig. S5c. The results revealed similar infection defects as observed in the *flot1-1* mutant. Collectively, these findings confirm the essential role of *Flot1* in rhizobial infection.

Regarding to *Flot1* RNAi, we observed an increased number of abnormal infection events in *Flot1* RNAi lines, which is a similar phenotype reported in *MtFlot4* RNAi (Haney and Long, 2010, PNAS). However, in the case of *Ljflot1-1*, we found that there were fewer infection foci and ITs, although the number of abnormal ITs was not significantly higher compared to the wild type. The only notable difference was a higher ratio of abnormal to total infection events, this could be a consequence of the lower total infection events in the mutant. We further validated this phenotype by conducting *Flot1* CRISPR/Cas9 knock-out experiments, as illustrated in the Fig. 5j. Moreover, we have concerns that the *Flot1* RNAi line may have affected other Flotillins, we decided to remove the *Flot1* RNAi results and revised the corresponding description in Line 249-253 to reflect this decision.

13. Fig. 7a: There are no proper specificity controls for the Y2H assay using another kinase domain. This should be emphasized as other studies failed to demonstrate a reliable interaction between symbiotic receptors and flotillins (e.g. Haney and Long, 2009; Haney et al., 2011).

Response: To address this concern, we examined the interaction between LYK10 and Flot1 using the Y2H and Co-IP techniques. Unfortunately, LYK10-CD-BK and Flot1-BK Y2H assay displayed strong self-activation in SD-LWH, it is difficult to determine whether Flot1 interacts with LYK10 using this method. However, Flot1 did not associate with LYK10 in the Co-IP experiment (This result has been included in Fig. 7b). In addition, Flot1 did not interact with LYK10-CD in SD-LWHA using Y2H assay. The Y2H results are provided below.

We are uncertain as to why previous studies have been unable to detect interactions between symbiotic receptors and Flotillins. In the two papers discussing MtFlotillin, the PNAS paper by Haney and Long (2010) does not include any reports on symbiotic receptors, or interactions between symbiotic receptors and Flotillins. However, in Haney et al. (2011, Plant Cell), they observed rhizobia-induced colocalization of LYK3 and Flot4 in inoculated root hairs. Although they did not provide conclusive results on whether LYK3 can interact with Flot4, they did discuss this issue: "Taken alone, these results suggest that FLOT4 may be interacting with the kinase domain of LYK3. However, we observed limited co-distribution of FLOT4 and LYK3 in the absence of bacteria. This observation could indicate that the hcl-1 protein itself has altered localization. Alternatively, FLOT4 and LYK3 may not directly interact, or limited interaction between LYK3 and FLOT4 may be sufficient to cause a change in FLOT4 distribution." As our research focuses on different proteins, it is difficult for us to provide further comments. However, we do discuss the differences between our study and the *M. truncatula* study in the Line 408-427 section.

14. Fig. 7b: Are the authors sure about the molecular weight indicated for FLOT1? Isn't much smaller compared to the NFRs?

Response: Yes, that's correct. The different molecular weight because of that Flot1 was tagged with

GFP, while NFR1/5 and RinRK1 were tagged with smaller tags (Myc or Flag). The molecular weight of each protein is shown below.

Flot1: 382 a.a. \approx 45.8 KD; Flot1-GFP: 45.8 KD + 27 KD=72.8 KD

NFR1: 622 a.a. \approx 74.6 KD; NFR1-Myc: 74.6 KD + 1.2 KD=75.8 KD

NFR5: 596 a.a. \approx 71.5 KD; NFR5-Myc: 71.5 KD + 1.2 KD=72.7 KD

RinRK1: 627 a.a. \approx 75.2 KD; RinRK1-3Flag: 75.2 KD + 3 KD=78.3 KD

15. line 312/Suppl. Fig. 9b: There are also punctate patterns with NFR5 in this panel. None of these observations is quantified. Why are these patterns not observed in Lotus root hairs?

Response: NFR5-Flot1 exhibited only a few punctate patterns, which were significantly fewer than RinRK1-Flot1 in *N.benthamiana* leaves. We confirmed the functionality of the BiFC constructs using *N. benthamiana*; however, our primary focus was to analyze the phenomena occurring before and after rhizobia inoculation. Consequently, we omitted the BiFC figures in *N. benthamiana* due to word limitation.

As we mentioned previously, we occasionally observed punctate localization in *Lotus* root hairs. We attribute this observation to either differences in expression levels, or limitations of the confocal system used. This phenomenon has been acknowledged and discussed, such as “Due to the spatiotemporal dynamics and size, microscopic visualization of membrane domains is rarely trivial (Konrad and Ott, 2017, Trends in Plant Science)”. We have included further discussion on this issue at Line 428-434.

16. Fig. 8: Why do the authors observe such as strong band shift for the mutated kinase domains? Could they please indicate the MW in the anti-His WB?

Response: We apologize for the lack of clear labelling in the previous version. We have made the necessary changes to address this issue. In Fig. 8a, we have replaced the Anti-His WB with a new image taken with a longer exposure time compared to the initial picture. Additionally, we have properly labelled panels in Fig. 8 a & b to indicate which protein each band corresponds to.

Regarding the anti-His WB, it is important to note that the upper strong band of NFR1-CD represents a phosphorylation form, as evidenced by its migration pattern. The weaker band below, which has the same molecular weight as the mutated kinase domains, corresponds to the non-

phosphorylated form of NFR1-CD.

Reviewer #1 (Remarks to the Author):

My concerns have been addressed by the authors.

Reviewer #2 (Remarks to the Author):

The revised version of the manuscript entitled "RinRK1 promotes accumulation of Nod Factor receptors in nanodomain enhancing rhizobial infection of legume roots" was improved to answer all my comments. The title is however overoptimistic as no experiment shows that accumulating Nod factor receptor complexes in nanodomain-like structures at the tip of root hairs ENHANCES rhizobial infection. So a more balanced title should be proposed such as "RinRK1 promotes the accumulation of Nod Factor receptor complexes in nanodomain-like structures at the tip of infected root hairs". Similarly, in the abstract, the last sentence should be changed as no experiment shows that the relocalisation AMPLIFIES the NF signaling, as well as l. 108 at the end of the introduction and l. 438 in the discussion (in both cases delete "enhance", and maybe just use "allow"). In this respect, the description l.174-175 is much more appropriate ("we speculate... tends to accumulate..."), as well as l. 467 ("may enhance").

Minor comments:

l.100, say clearly that RinRK1 has no functional kinase domain, with the appropriate reference showing this.

l. 120, it remains still unclear to me why MtLYK10 was used as a control and not LjEPR3. Please better justify

l.127 should be moved to l. 122: "... between MtLYK10 and RinRK1, despite the protein expression in yeast was confirmed by western blot analysis (Fig. 1b and Supplementary Fig.1), making it suitable as...

l. 142, "A split-luciferase complementation (...), further confirming that , can interact with..." (delete "indeed" here)

l. 167, here and elsewhere, replace "it's" by "it is"; and l. 422, "we're" by "we are"

l. 170, delete "with"

l. 176, "to determine IF"; l. 177 NFs; l. 178 brackets are useless; l. 180 "barley" should be "barely"!

l.187, replace "crucial" by "required"

l. 189 and l.289, delete "were analyzed using ImageJ", and l. 189 replace "confirming" by "confirmed". If you want, you can add at the end of the sentence l. 189 "(see details in methods)" but to me this is not needed.

"ITs" should be defined l.71, not l. 247

l.249, "To confirm THAT..."

l. 260, "THE flot1-1..."

l. 274, delete "consistent with RinRK1", or shortly clarify what do you mean

l. 276, "THE accumulation"

l. 300, move "ratio" at the end of the sentence ("tip and trunk RATIO").

l. 309, "this reduced" instead of "this diminished"

l. 325 "assays"

l. 353, you may start saying that "Earlier studies demonstrated that NFR1, in contrast to NFR5 and RinRK1, has serine/threonine...", and adding the appropriate references.

The sentence l. 371-373 could be moved after the next sentence (l. 376); "where" is missing before "an essential residue". In the Fig 8a, why bands of different size are detected, as only point mutations were generated compared to the WT NFR1 CD?

l. 386-389, the sentence could be improved to clarify that NFR1 T481A behaves differently than NFR1Y429F, being not anymore responsive to M. loti, by saying something like "in contrast to NFR1T481A, which root tip relocalisation is not anymore observed in response to M. loti,..."

l. 391-393, it is somewhat strange to finish the result section by a negative sentence.

l. 403, delete "spatiotemporal" as this is not defined here, or alternatively bring here the sentence from l. 428-429 about the M. loti/NF root tip relocalisation.

l. 407-408 are useless and should be deleted. Directly start this paragraph by "In addition, Flot1 interacts with NFR1, NFR5 and RinRK1". You could shortly remind here what is Flot1 presumed function here.

l. 420, "contributed" should be "related"
l. 423 "is " is missing before " a homologue"
l. 427, you could add a sentence saying that "further studies would be needed to clarify the relative role of different flotillins in *M. truncatula* vs *L. japonicus*"

l. 435, "A MODEL WHERE RinRK1 COULD BE a scaffold protein"
l. 452, delete "it" and add "phosphorylation" before "activity", and "on the Flot1 Tyr residue" instead of "against"
l. 453 ", OR THAT other unidentified kinases..."
l. 456, delete "has a regulatory cycle" or shortly explain what do you mean.
l. 458, replace "conclude" by "propose"
l. 462, "ALREADY present in nanodomains"
l. 463 is not clear as you mention here that NF "leads to the production of RinRK1 and Flot1", whereas the sentence just before you said that they are already present. Please clarify.
l.464, RinRK1 WOULD then serve.... as a scaffold TO facilitate ..., LEADING to the recruitment..."
l. 466-469, this last sentence should be improved as currently it is too speculative and not very clear.
Please similarly double check for little English mistakes in the Methods and Figure/Supp material legend sections.

Reviewer #4 (Remarks to the Author):

Zhou and co-authors studied the function of a cell surface receptor, RinRK1, in regulating legume-rhizobial symbiosis in *Lotus Japonicus*. The authors propose that RinRK1 is organized into plasma membrane nanodomains and function as a protein scaffold regulating the association of the Nod factor receptors NFR1 and NFR5 with a membrane protein scaffold, FLOT1, thereby regulating their nanodomain organization and activity.

The authors do not provide substantial evidences supporting these claims. None of the experiments assessed the organization of Nod factor receptors in nanodomains nor of the potential function of RinRK1 in this context. The proposed scaffolding function of NFR1 solely rely on one Co-IP experiment performed upon transient expression in a heterologous system. The proposed nanodomain organization of RinRK1 primarily rely on its detection in detergent resistant membranes upon transient expression in a heterologous system.

The data presented by the authors show that RinRK1 associates with both NFR1 and NFR5. The authors observed that inoculation with rhizobia or treatment with Nod Factor promotes the polar localization of RinRK1, NFR1 and NFR5 at the tip of root hairs. The authors observed that conditional polar localization of NFR1 and NFR5 is RinRK1- and FLOT1-dependent. Based on the current set of data, I would suggest the authors to articulate the main claims of their manuscript around these points, which provide a step forward in the functional characterization of RinRK1 in regulating symbiosis.

Please find below specific comments:

1. Throughout the manuscript: the authors should not describe polar localization of studied cell surface receptors as nanodomain organization. Polar localization and presence in detergent resistance membrane fraction do not constitute proxies for nanodomain organization.
2. The quantitative analysis of fluorescence ratio tip/trunk present few data points compared to the number indicated on each image (Figure 2, 3 4 and 6). It is unclear whether the scoring presented on each image is based on quantification of fluorescence intensity, or whether the quantification of fluorescence intensity has been performed on selected root hair based on qualitative assessment of the polar localization of the receptors. It would be best to present the quantification the fluorescence ratio for all imaged root hairs.
3. Detergent resistant membranes cannot be used synonymously to plasma membrane

nanodomain. Therefore, the western blot of figure 2a cannot be annotated ND (nanodomain). This experiment is not properly designed to assess enrichment of RinRK1-eGFP in this biochemical fraction. One would need to compare relative abundance of RinRK1-eGFP in plasma membrane fractions and DRM fractions compared to total PM/DRM protein (or targeted control proteins). I would suggest here to simply rephrase the text, I do not think that such comparative biochemical is relevant here. Finally, it would be best to include controls to first characterise the isolated biochemical fraction. I stress here again that detergent resistant membranes can not be consider as a proxy for plasma membrane nanodomain organization.

4. Figure 7b: The tested protein-protein interactions and corresponding control should be presented on the same western blot. Performed as presented here, the experiment is difficult to interpret.

5. Figure 7b: The expression of RinRK1-Flag seems to decrease the accumulation of Flot1-eGFP, was this also observed during confocal microscopy experiments (Supplemental figure 6d)? This observation may be relevant to understand the function of RinRK in regulating symbiosis.

6. L195-196: it is stated here that *M. loti* stimulates the interaction between RinRK1 and NFR1 and NFR5. The data show changes in the subcellular localization of the BiFC signal, which correlates with changes in the receptor's accumulation at subcellular level as presented in other figures (e.g. Figure 3), not a change in protein-protein interactions themselves.

Similarly, the absence of changes in the ratio of fluorescence (root hair tip/trunk), cannot be interpreted as changes in the association between FLOT1 with NFR1 and NFR5 (Figure 7, L345-347) but simply reflect absence of changes in protein accumulation at subcellular level.

7. The localization of RinRK1 and FLOT1 shown in supplemental figure 6d seems peculiar compared to their localization in supplemental figure 6a using the same experimental set up and compared to their localization when expressed in root hair (in all other figures). In any cases, these observations (Sup. figure 6d) do not provide strong bases to describe RinRK1 as being nanodomain organized. To assess potential nanodomain organization of RinRK1, the authors should analyze cell surfaces, in which uneven distribution within a membrane plane could potentially be resolved.

8. There is no control for the BiFC experiments presented in Figure 4 and Figure 7.

9. There is no appropriate control (e.g. another plasma membrane-localized protein) to assess the interaction of FLOT1 with tested receptor kinases in Y2H (Figure 7a) shown in the manuscript.

10. Figure 8a and 8b, L361-365. The experiments are not properly designed to test NFR1 Tyrosine kinase activity. It is good practice to include kinase dead version of kinases in such assays to exclude transphosphorylation by potential contaminant *E. coli* kinase(s). The T481A mutation is located in the activation loop of NFR1 kinase domain and has been shown to abolish its kinase activity (Madsen et al., *Plant Journal* 2011). In consequence, the data presented by the authors suggest that phosphorylation of NFR1 Y429 residue is not due to NFR1 auto-phosphorylation but to trans-phosphorylation of NFR1 by contaminating protein kinases.

11. Nanodomain-organized should be used instead of nanodomain-associated (e.g L21). Nanodomain associated implies the association of FLOT1 with pre-existing membrane structure.

12. L26-27: The authors do not assess NF-induced signaling (i.e monitoring of molecular marks induced by NF perception), please rephrase.

REVIEWER COMMENTS

Reviewer #1 (Remarks to the Author):

My concerns have been addressed by the authors.

Response: Thanks a lot to this reviewer for supporting our research.

Reviewer #2 (Remarks to the Author):

The revised version of the manuscript entitled “RinRK1 promotes accumulation of Nod Factor receptors in nanodomain enhancing rhizobial infection of legume roots” was improved to answer all my comments. The title is however overoptimistic as no experiment shows that accumulating Nod factor receptor complexes in nanodomain-like structures at the tip of root hairs ENHANCES rhizobial infection. So a more balanced title should be proposed such as “RinRK1 promotes the accumulation of Nod Factor receptor complexes in nanodomain-like structures at the tip of infected root hairs”. Similarly, in the abstract, the last sentence should be changed as no experiment shows that the relocalisation AMPLIFIES the NF signaling, as well as l. 108 at the end of the introduction and l. 438 in the discussion (in both cases delete “enhance”, and maybe just use “allow”). In this respect, the description l.174-175 is much more appropriate (“we speculate... tends to accumulate...”), as well as l. 467 (“may enhance”).

Response: Acknowledging the valuable feedback from this reviewer, and incorporating the comments from Reviewer 3, we have revised the title of our study to “RinRK1 promotes accumulation of Nod Factor receptors in nanodomain-like structures at the tip of infected root hairs”.

Additionally, we have optimized the relevant sentences in the Abstract, Results, and Discussion sections to align with the revised title.

Minor comments:

l.100, say clearly that RinRK1 has no functional kinase domain, with the appropriate reference showing this.

Response: Apologies for the confusion. We have now added the reference and clarified this aspect.

l. 120, it remains still unclear to me why MtLYK10 was used as a control and not LjEPR3. Please better justify

Response: We apologize for the confusion caused. Our objective is to study the interaction between RinRK1 and NF receptors. To serve as an appropriate control, we chose MtLYK10, which is a transmembrane LysM RLK. We have multiple constructs related to MtLYK10, and one student in my lab is specifically working on this gene.

Initially, when researching this gene, it was referred to as EPR3 in *L. japonicus*, thus we used MtEPR3 at that time. However, a study conducted on *M. truncatula* named this gene as MtLYK10 (Mailet et al., 2020, Plant J. doi: 10.1111/tbj.14625). Therefore, in this manuscript, we are using the name MtLYK10 instead of EPR3 to align with the *M. truncatula* study.

In the Fig 8a, why bands of different size are detected, as only point mutations were generated

compared to the WT NFR1 CD?

Response: Apologies for the confusion. The upper strong band observed in the gel represents a phosphorylated form of NFR1-CD, as evidenced by its distinct migration pattern. Conversely, the weaker band appearing below, which shares the same molecular weight as the mutated kinase domains, corresponds to the non-phosphorylated form of NFR1-CD. To provide clarity, we have labelled these bands accordingly.

l. 463 is not clear as you mention here that NF “leads to the production of RinRK1 and Flot1”, whereas the sentence just before you said that they are already present. Please clarify.

Response: Apologies for the confusion that may have arisen. When we mentioned that “leads to the production of RinRK1 and Flot1”, we mean that it results in the transcriptional upregulation of *RinRK1* and *Flot1*. We have now clarified this to avoid any misunderstandings.

l. 466-469, this last sentence should be improved as currently it is too speculative and not very clear.

Please similarly double check for little English mistakes in the Methods and Figure/Supp material legend sections.

Response: Thank you very much for the thorough reviewer’s suggestion. We have carefully revised and double-check the English language used in the text, and we also had a native speaker edit it for further improvement. We appreciate your attention to detail and strive to provide the best quality in our manuscript.

Reviewer #4 (Remarks to the Author):

Zhou and co-authors studied the function of a cell surface receptor, RinRK1, in regulating legume-rhizobial symbiosis in *Lotus Japonicus*. The authors propose that RinRK1 is organized into plasma membrane nanodomains and function as a protein scaffold regulating the association of the Nod factor receptors NFR1 and NFR5 with a membrane protein scaffold, FLOT1, thereby regulating their nanodomain organization and activity.

The authors do not provide substantial evidences supporting these claims. None of the experiments assessed the organization of Nod factor receptors in nanodomains nor of the potential function of RinRK1 in this context. The proposed scaffolding function of NFR1(RinRK1) solely rely on one Co-IP experiment performed upon transient expression in a heterologous system. The proposed nanodomain organization of RinRK1 primarily rely on its detection in detergent resistant membranes upon transient expression in a heterologous system.

Response: In response to this criticism, we would like to present several aspects that support our claims:

1) Confocal microscopy is widely used to observe protein organized in nanodomains or microdomains, which often exhibit distinct punctate localization. Examples of such proteins include flotillin and various receptors (Li et al. 2012. *Plant Cell* 24, 2105–2122. Haney et al. 2010. *PNAS* 107, 478-483. Haney et al. 2011. *Plant Cell* 23, 2774-2787). We occasionally detected punctate signals corresponding to NFR1 or NFR5 in root hair tips, either individually

or in NFR1-RinRK1 or NFR5-RinRK1 in *L. japonicus* root hairs (as shown in Fig. S3c & S5 and below). We think that the reason that punctate signals are only observed in our experiments at a low frequency is likely due to low expression levels. Notably, the accumulation of fluorescent signals at the root hair tip after rhizobial or NF inoculation is consistent with previous publications on nanodomain proteins such as MtFlot4, GmFlot2/4 and GmFWL1 (Haney and Long, PNAS, 2010; Qiao et al., Plant Cell and Enviro. 2017; typical images were shown in below).

Fig. 3. FLOTs localize to membrane-associated puncta and become polar after inoculation. We generated A17 hairy roots expressing 35S:FLOT2:GFP or FLOT4:GFP driven by its native promoter. Transgenic roots were visualized by using a spinning-disk confocal microscope. (Scale bars: 15 μ m.) FLOT4:GFP and FLOT2:GFP are visibly punctate and evenly distributed in the membranes of uninoculated root hair cells. FLOT4:GFP puncta localize to root hair tips by 1 dpi, whereas the localization of FLOT2:GFP does not change upon inoculation. FLOT2:GFP is weakly polar in uninoculated epidermal cells (arrows) and remains polar on inoculation. At least 15 transgenic lines were observed for each treatment. Representative images are shown. Root hair images are overlays of 100 sections, taken at 0.2- μ m increments.

Figure 5. Subcellular localization of mCherry-GmFLOT2/4 fusion protein in soybean root hair cell in response to *Bradyrhizobium japonicum* inoculation. Soybean root hair cells were mock-inoculated (a, b, e and f) or inoculated with *B. japonicum* (c, d, g, h and i). One day (a–d) and 7 d (e–i) after inoculation, the transgenic root hair cells were observed under an epifluorescent confocal microscope. Compared to mock-inoculated conditions and similarly to GmFWL1 (Fig. 5), GmFLOT2/4 accumulation at the tip of the root hair cells in response to *B. japonicum* inoculation was observed as soon as 24 h after inoculation and continued 7 d after inoculation. White arrows highlight the accumulation of the mCherry-GmFLOT2/4 protein at the tip of the root hair cells upon *B. japonicum* inoculation. Supporting Information Fig. S6 shows the quantification of the mCherry signal in inoculated and mock-inoculated transgenic root hair cells.

- 2) While detergent-resistant membranes (DRMs) and M β CD may not fully represent nanodomain proteins, they are commonly used methods for studying lipid-protein targeting in nanodomains. Both RinRK1 and Flot1 were found to be enriched in DRMs, and simultaneous treatment with M β CD and *M. loti* resulted in reduced accumulation of these proteins at root hair tips in response to NFs.
- 3) Flotillin is a well-established marker for nanodomains in mammal and several plant species. Based on our confocal microscopy observations, the presence of RinRK1, NFR1, NFR5 and Flot1 in nanodomain-like structures at the tips of root hairs in response to rhizobia or NFs is supported by the evidence obtained from DRMs, M β CD treatment, and similarities observed with Flot1, a widely recognized marker protein for membrane nanodomains.

In addition to the Co-IP experiment conducted in *N. benthamiana*, we employed various approaches to investigate the proposed scaffolding function of RinRK1. These include examining the accumulation of NFR1/NFR5/Flot1 alone or BiFC constructs (NFR1/NFR5-Flot1) at the tips of *L. japonicus* root hairs following rhizobia inoculation, but this was significantly reduced in *rinrk1* mutant (Fig. 3, 6, & 7).

Considering the feedback from the reviewers, we have revised our conclusion and refrained from using the term “scaffold” for RinRK1. Throughout the manuscript, including the title and abstract, we incorporate the term “nanodomain-like structures” to provide a more cautious interpretation of our findings.

The data presented by the authors show that RinRK1 associates with both NFR1 and NFR5. The authors observed that inoculation with rhizobia or treatment with Nod Factor promotes the polar localization of RinRK1, NFR1 and NFR5 at the tip of root hairs. The authors observed that conditional polar localization of NFR1 and NFR5 is RinRK1- and FLOT1-dependent. Based on the current set of data, I would suggest the authors to articulate the main claims of their manuscript around these points, which provide a step forward in the functional characterization of

RinRK1 in regulating symbiosis.

Response: Thank you for your suggestion. Taking the reviewers' comments into consideration, we have made appropriate adjustments to our claims regarding the role of RinRK1 in the rhizobial infection process. We hope these revisions address the concerns raised and meet your expectations.

Please find below specific comments:

1. Throughout the manuscript: the authors should not describe polar localization of studied cell surface receptors as nanodomain organization. Polar localization and presence in detergent resistance membrane fraction do not constitute proxies for nanodomain organization.

Response: Point taken. Based on our microscopic observation and other experimental findings, we have taken into account the reviewers' feedback and replaced "nanodomain" with "nanodomain-like structures" throughout our research to provide a more cautious interpretation of our results.

2. The quantitative analysis of fluorescence ratio tip/trunk present few data points compared to the number indicated on each image (Figure 2, 3 4 and 6). It is unclear whether the scoring presented on each image is based on quantification of fluorescence intensity, or whether the quantification of fluorescence intensity has been performed on selected root hair based on qualitative assessment of the polar localization of the receptors. It would be best to present the quantification the fluorescence ratio for all imaged root hairs.

Response: We apologize for any confusion caused by our previous explanation of the inflorescence ratio. To clarify, the fluorescence intensity scoring was conducted on selected root hairs, taking into account the qualitative assessment of the polar localization of the receptors. While an ideal scenario would involve quantifying the fluorescence ratio for all imaged root hairs, it should be noted that normally not all root hairs respond to rhizobia/NF inoculation, such as through root hair deformation or infection.

In our study, all the numbers presented in the figure panels were obtained by combining data from two or three experimental replicates. The fluorescence ratio between root hair tips and the trunks was analyzed using at least 15 individual root hairs from different transgenic plants. We have included this clarification in the Methods section (Line 690-715) to provide clear and accurate information.

3. Detergent resistant membranes cannot be used synonymously to plasma membrane nanodomain. Therefore, the western blot of figure 2a cannot be annotated ND (nanodomain). This experiment is not properly designed to assess enrichment of RinRK1-eGFP in this biochemical fraction. One would need to compare relative abundance of RinRK1-eGFP in plasma membrane fractions and DRM fractions compared to total PM/DRM protein (or targeted control proteins). I would suggest here to simply rephrase the text, I do not think that such comparative biochemical is relevant here. Finally, it would be best to include controls to first characterise the isolated biochemical fraction. I stress here again that detergent resistant membranes can not be consider as a proxy for plasma membrane nanodomain organization.

Response: We acknowledge and understand the reviewer's point that DRM (detergent resistant

membranes) alone does not definitively confirm the presence of nanodomain proteins. However, it is worth noting that Flotillin has been widely recognized as a reliable marker for nanodomains in both mammal and plant species. As shown in Fig S8c, the data demonstrate Flot1's presence as a control for DRM, and also reveal the enrichment of RinRK1 within the DRMs. In response to this feedback, we have made the necessary adjustments by replacing “ND” with “DRM” in Fig 2a and Fig. S8c, and the relevant sections of the manuscript to accurately reflect these findings.

4. Figure 7b: The tested protein-protein interactions and corresponding control should be presented on the same western blot. Performed as presented here, the experiment is difficult to interpret.

Response: Thank you for your suggestion. We have made the appropriate changes to the presentation of the Co-IP results in Figure 7. The results for the tested protein-protein interactions and the corresponding controls are now depicted in Fig 7b, while RinRK1 enhancing Flot1 interaction with NFR1/NFR5 is shown in Fig. 7c.

5. Figure 7b: The expression of RinRK1-Flag seems to decrease the accumulation of Flot1-eGFP, was this also observed during confocal microscopy experiments (Supplemental figure 6d)? This observation may be relevant to understand the function of RinRK in regulating symbiosis.

Response: No, not really. We did not observe any significant changes in Flot1 levels in relation to the expression of RinRK1.

6. L195-196: it is stated here that *M. loti* stimulates the interaction between RinRK1 and NFR1 and NFR5. The data show changes in the subcellular localization of the BiFC signal, which correlates with changes in the receptor's accumulation at subcellular level as presented in other figures (e.g. Figure 3), not a change in protein-protein interactions themselves. Similarly, the absence of changes in the ratio of fluorescence (root hair tip/trunk), cannot be interpreted as changes in the association between FLOT1 with NFR1 and NFR5 (Figure 7, L345-347) but simply reflect absence of changes in protein accumulation at subcellular level.

Response: Apologies for any confusion caused. BiFC is indeed a technique used to visualize protein-protein interactions directly in living cells. However, it's important to note that BiFC methods only determine the close physical proximity of two fluorophore-tagged fusion proteins *in vivo*. The interpretation of such a tight contact as conclusive evidence of a true protein-protein interaction may be subject to debate. In order to accurately reflect our findings, we have revised the statement to “NFs promote the accumulation of NFR1 and NFR5 at the root-hair tip”. Additionally, we have removed the claim in L345-347. We apologize for any previous inaccuracies and appreciate your clarification.

7. The localization of RinRK1 and FLOT1 shown in supplemental figure 6d seems peculiar compared to their localization in supplemental figure 6a using the same experimental set up and compared to their localization when expressed in root hair (in all other figures). In any cases, these observations (Sup. figure 6d) do not provide strong bases to describe RinRK1 as being nanodomain organized. To assess potential nanodomain organization of RinRK1, the authors should analyze cell surfaces, in which uneven distribution within a membrane plane could potentially be resolved.

Response: Thank you for your suggestion. We have added z-stack images in Fig. S8d.

8. There is no control for the BiFC experiments presented in Figure 4 and Figure 7.

Response: Thank you for your suggestion. To validate the interaction, we performed BiFC assays using RinRK1-LYK10 or Flot1-LYK10 constructs as negative controls in *N. benthamiana*. Despite confirming the protein expression using WB analysis, the interactions of RinRK1-MtLYK10 or Flot1-MtLYK10 on the PM were hardly detectable. The correspond results of the BiFC and WB analysis are now included in Fig. S2 and Fig. S9.

9. There is no appropriate control (e.g. another plasma membrane-localized protein) to assess the interaction of FLOT1 with tested receptor kinases in Y2H (Figure 7a) shown in the manuscript.

Response: To address this concern, we investigated the interaction between LYK10 and Flot1 using the Y2H assay. Our results indicated that Flot1 does not interact with LYK10-CD, as evidenced by the lack of yeast growth on selective medium (SD-LWHA), although the protein expression was validated by WB. These data were shown in Fig. S9 now.

10. Figure 8a and 8b, L361-365. The experiments are not properly designed to test NFR1 Tyrosine kinase activity. It is good practice to include kinase dead version of kinases in such assays to exclude transphosphorylation by potential contaminant *E. coli* kinase(s). The T481A mutation is located in the activation loop of NFR1 kinase domain and has been shown to abolish its kinase activity (Madsen et al., Plant Journal 2011). In consequence, the data presented by the authors suggest that phosphorylation of NFR1 Y429 residue is not due to NFR1 auto-phosphorylation but to trans-phosphorylation of NFR1 by contaminating protein kinases.

Response: To determine the Tyr phosphorylation activity of NFR1, we expressed and purified kinase domain of NFR1 (NFR1-CD). To assess its Tyr phosphorylation status, we employed an anti-phosphotyrosine (pTyr) antibody. Additionally, we used an anti-phosphoserine/threonine (pSer/Thr) antibody to evaluate the NFR1 Ser/Thr phosphorylation status. As a control, we utilized NFR1^{T481A} which was used in previous study (Madsen et al., 2011). Furthermore, we conducted the expression and purification of both NFR1 and its mutant variants (NFR1^{Y429F} and NFR1^{T481A}) from *E. coli*. When comparing the phosphorylation levels using pTyr and pSer/Thr antibodies, we observed a significant reduction in the phosphorylation bands of both NFR1^{Y429F} and NFR1^{T481A} compared to the wild-type (WT) NFR1. This finding provides strong evidence that the Y429 residue in NFR1 is responsible for the reduction in Tyr phosphorylation and not due to trans-phosphorylation by contaminating protein kinases.

Based on our *in vitro* phosphorylation assays, we found that NFR1^{Y429F}-CD, which corresponds to Y428 residue of AtCERK1, lost its ability to undergo Ser/Thr phosphorylation. Furthermore, its capacity for Tyr phosphorylation was significantly diminished. This suggested that the phosphorylation of NFR1 Y429 residue predominantly occurs as a result of NFR1 auto-phosphorylation, as observed in previous studies involving AtCERK1 autophosphorylation (Liu, J. et al. 2017. *Cell Host Microbe* 23, 241-253).

11. Nanodomain-organized should be used instead of nanodomain-associated (e.g L21). Nanodomain associated implies the association of FLOT1 with pre-existing membrane structure.

Response: Point taken. Thank you for your suggestion, we have replaced “nanodomain-associated”

with “nanodomain-organized”.

12. L26-27: The authors do not assess NF-induced signaling (i.e monitoring of molecular marks induced by NF perception), please rephrase.

Response: Thank you for your suggestion. We have rephrased the sentence.

Reviewer #4 (Remarks to the Author):

In their revised manuscript entitled "RinRK1 Promotes Accumulation of Nod Factor Receptors in Nanodomain-like structures at the tip of infected root hairs" the authors answered my previous concerns.

Here are minor comments on the revised manuscript:

-The figure panel 7b would need to be double checked by the authors as it is not consistent with a previous version of the figure and the statement made by the authors. In this figure panel, the co-immunoprecipitation experiment doesn't show that expression of RinRK1 promotes the association between Flot1 and NFR1 or NFR5. The values below the blot do not reflect relative variation of co-immunoprecipitated Flot1-eGFP. For instance, while more NFR5-myc is immunoprecipitated less Flot1-eGFP is co-immunoprecipitated (lane 3 and 4 respectively), indicating that expression of RinRK1 decrease the association of NFR5 and Flot1-eGFP.

-It is unclear whether the punctate structure occasionally observed and presented in Fig. S3c & S5 are plasma membrane domains or intracellular punctate indicative of intracellular trafficking events. Presenting time laps imaging could clarify this point.

-It should be noted that authors do not assess enriched in DRM but show that RinRK1-eGFP is detected in this fraction. I do not think that comparative analysis of RinRK1-eGFP relative abundance in DRM is pertinent and I would suggest the authors to revise the text to reflect that the presence of RinRK1-eGFP in DRM was tested, not its enrichment.

-methyl- β cyclodextrin (M β CD) should not be described as an inhibitor, but rather be described as a chelator extracting sterols from membrane (L172).

REVIEWER COMMENTS

Reviewer #4 (Remarks to the Author):

In their revised manuscript entitled “RinRK1 Promotes Accumulation of Nod Factor Receptors in Nanodomain-like structures at the tip of infected root hairs” the authors answered my previous concerns.

Here are minor comments on the revised manuscript:

-The figure panel 7b would need to be double checked by the authors as it is not consistent with a previous version of the figure and the statement made by the authors. In this figure panel, the co-immunoprecipitation experiment doesn't show that expression of RinRK1 promotes the association between Flot1 and NFR1 or NFR5. The values below the blot do not reflect relative variation of co-immunoprecipitated Flot1-eGFP. For instance, while more NFR5-myc is immunoprecipitated less Flot1-eGFP is co-immunoprecipitated (lane 3 and 4 respectively), indicating that expression of RinRK1 decrease the association of NFR5 and Flot1-eGFP.

Response: We apologize for the confusion caused. Co-IP experiments were conducted with anti-GFP agarose using total extracts, and immunoblotting was performed utilizing anti-Myc or anti-GFP antibody. The values presented below the blot correspond to the relative quantities of NFR1-Myc or NFR5-Myc co-immunoprecipitated with Flot1-eGFP in the absence (lane 1 and 3) or presence (lane 2 and 4) of RinRK1. These values were normalized by comparing lane 2 to lane 1, and lane 4 to lane 3.

For lane 3, it is noteworthy that Flot1-eGFP is immunoprecipitated to a greater extent compared to lane 4 (plus RinRK1), whereas NFR5-Myc exhibits higher levels in lane 4. This suggests that expression of RinRK1 can enhance the association of NFR5 and Flot1. The legend of Figure 7c has been revised accordingly.

-It is unclear whether the punctate structure occasionally observed and presented in Fig. S3c & S5 are plasma membrane domains or intracellular punctate indicative of intracellular trafficking events. Presenting time laps imaging could clarify this point.

Response: We appreciate the reviewer's observation and acknowledge the point regarding the punctate structure showed in Figure S3c or S5, which may potentially represent plasma membrane nanodomain structures or intracellular trafficking events. Previous studies have showed that plant structural sterols are present within endocytic networks and play a role in a constitutive endocytic cycling of specific plasma membrane proteins (The endocytic network in plants. <https://doi.org/10.1016/j.tcb.2005.06.006>). Additionally, membrane microdomain-associated endocytosis has been defined as clathrin-independent endocytic pathway (A membrane microdomain-associated protein, *Arabidopsis* Flot1, is involved in a clathrin-independent endocytic pathway and is required for seedling development. <https://doi.org/10.1105/tpc.112.095695>; Sterols regulate endocytic pathways during flg22-induced defense responses in *Arabidopsis*. <https://doi.org/10.1242/dev.165688>). For instance, it has been shown that BRI1 localizes at membrane microdomains and undergoes internalization through membrane microdomain associated endocytosis (Spatiotemporal dynamics of the BRI1 receptor and its regulation by membrane

microdomains in living *Arabidopsis* cells. <https://doi.org/10.1016/j.molp.2015.04.005>). Therefore, proteins localized to membrane microdomains are closely associated with intracellular trafficking events. However, since the punctate structures were occasionally observed in our study, we did not stain the endosomal marker dye FM4-64 or conducted time-course experiments, we cannot exclude the possibility that the punctate structures could represent an endosomal compartment. We will investigate this phenomenon in future.

-It should be noted that authors do not assess enriched in DRM but show that RinRK1-eGFP is detected in this fraction. I do not think that comparative analysis of RinRK1-eGFP relative abundance in DRM is pertinent and I would suggest the authors to revise the text to reflect that the presence of RinRK1-eGFP in DRM was tested, not its enrichment.

Response: Thank you for your suggestion and we have revised the text (Line 158).

-methyl- β cyclodextrin (M β CD) should not be described as an inhibitor, but rather be described as a extracting sterols from membrane (L172).

Response: Point taken. We have replaced with “a compound known for its ability to deplete sterols and disrupt nanodomains” according to the function of M β CD.

Reviewer #4 (Remarks to the Author):

The authors answered my previous concerns and amended the manuscript accordingly.